# Trends, barriers and enablers to measles immunisation coverage in Saskatchewan, Canada: A mixed methods study

**Marcus M. Ilesanmi**[1]*, **Sylvia Abonyi**[1,2], **Punam Pahwa**[1,3], **Volker Gerdts**[4,5], **Michael Scwandt**[6,7], **Cordell Neudorf**[1,8]

**1** Department of Community Health and Epidemiology, College of Medicine, University of Saskatchewan, Saskatoon, SK, Canada, **2** Saskatchewan Population Health and Evaluation Research Unit (SPHERU), University of Saskatchewan, Saskatoon, SK, Canada, **3** Canadian Centre for Health and Safety in Agriculture, University of Saskatchewan, Saskatoon, SK, Canada, **4** Vaccine and Infectious Disease Organization-International Vaccine Centre (VIDO-InterVac), University of Saskatchewan, Saskatoon, SK, Canada, **5** Department of Veterinary Microbiology, Western College of Veterinary Medicine, University of Saskatchewan, Saskatoon, SK, Canada, **6** School of Population and Public Health, University of British Columbia, Vancouver, BC, Canada, **7** Vancouver Coastal Health, Office of the Chief Medical Health Officer, Vancouver, BC, Canada, **8** Health Surveillance & Reporting, Saskatchewan Health Authority (SHA), Saskatoon, SK, Canada

* marcus.ilesanmi@usask.ca

**Data Availability Statement:** Data cannot be shared publicly because of confidential issues. The data set used for the study is owned and in the custody of a third-party organization – the

## Abstract

Many social, cultural, and systemic challenges affect the uptake of measles immunisation services. Prior studies have looked at the caregivers' perspectives, but little is known about the perspectives of the health care providers on the barriers of measles immunisation services in Canada. This study examined measles immunisation coverage trends across the regional health authorities in Saskatchewan and explored the barriers and enablers to measles immunisation coverage from providers' perspectives. The study adopted an explanatory sequential mixed method. We utilized the entire population of 16,582 children under two years of age available in the Saskatchewan Immunisation Management System (SIMS) registry for 2002 and 2013 in aggregate format and interviewed 18 key informants in pre-determined two-stages in 2016 and 2017. The quantitative analysis was done with Joinpoint regression modelling, while the qualitative interview data was analyzed using hybrid inductive and deductive thematic approaches. There was a 16.89%-point increase in measles immunisation coverage in the province from 56.32% to 73.21% between 2002 and 2013. There was also a persistently higher coverage among the affluent (66.95% - 82.37%) than the most deprived individuals (45.79% - 62.60%) in the study period. The annual rate of coverage change was marginally higher among the most deprived (16.81%; and average annual percentage change (AAPC) 2.0, 95% CI 1.7–2.2) than among the affluent group (15.42% and AAPC 3.0; 95% CI 2.0–4.0). While access-related issues, caregivers' fears, hesitancy, anti-vaccination challenges, and resource limitations were barriers to immunisation, improving community engagement, service delivery flexibility, targeted social responses and increasing media role were found useful to address the uptake of measles and other vaccine-preventable diseases immunisation. There is low coverage and inequity in measles immunisation uptake in Saskatchewan from social and institutional barriers.

Saskatchewan provincial government. There are ethical restrictions on sharing the data of the Ministry of Health. Researchers who wish to carry out similar studies are expected to apply to the provincial Ministry of Health (Population Health Branch) for approval as the legal custodian of the immunization registry data which is currently hosted in Panorama. For intending researchers, the data may be requested through: https://www.ehealthsask.ca/health-data/analytics/Pages/Researcher-Access-to-Data.aspx The requisition form can be found at the link: http://www.ehealthsask.ca/forms/Forms/RequestForInformationForm.pdf while the requisition should be mailed to: servicedesk@ehealthsask.ca No special privileges were received by the authors in accessing the data from the third-party organization that other researchers would not have.

**Funding:** The author(s) received no specific funding for this work.

**Competing interests:** The authors have declared that no competing interests exist.

**Abbreviations:** AAPC, Average Annual Percent Change; APC, Annual Percentage Change; HIT, Herd Immunity Threshold; LSE, Low Socioeconomic; MHO, Medical Health Officer; NACI, National Advisory Committee on Immunisation; RHA, Regional Health Authority; SDoH, Social Determinants of Health; S-EDQ, Socio-economic Deprivation Quintile; SIMS, Saskatchewan Immunisation Management System.

Even though there is evidence of disparity reduction among the different groups, the barriers to increasing measles immunisation coverage have implications for the health of the socio-economically deprived groups, the healthcare system and other vaccination programs. There is a need to improve policy framework for community engagement, targeted programs, and public health discourse.

## Introduction

Core to reducing the economic impact of diseases both on the health care financing and for children well-being is disease prevention achieved through immunisation, which has proved to be useful in the control and elimination of life-threatening infectious diseases [1, 2]. Among several infectious diseases, measles has received prominent attention internationally due to its high rate of infectivity [3–5]. Measles immunisation averts between 2 and 3 million deaths globally each year [2, 6]. Between 2000 and 2018, 23.2 million deaths were prevented [7] resulting in approximately 73% drop in measles cases within that period. With all the gains, however, in 2019, there were 869 770 measles cases worldwide, being the highest number since 1996 and with about 207 500 (23.9%) lost lives in the same year [8]. Despite all the efforts to improve measles immunisation coverage, the proportion of immunised population with measles antigen containing vaccine (MCV) remains under the 95 percent herd immunity threshold (HIT) [8]. A high percentage of unimmunized individuals portends high susceptibility and infectivity for vulnerable groups with unplanned public health expenditure and negative outcomes [9].

Measles is a notifiable disease to be reported by health care providers in Canada as recommended in the national guideline since 1924 [10]. The National Advisory Committee on Immunisation (NACI) emphasized the importance of elimination of indigenous measles since 1980 [11, 12], an objective which was reinforced by the Canadian Paediatric Association [13]. In 1994, Health Canada joined other Pan American Ministries of Health to set year 2000 measles elimination target in the Western Hemisphere [11, 14, 15] Measles elimination was however achieved in 1998 when Canada was declared free of endemic measles infection [**11, 15**]. Canada adopts 2-dose measles antigen-containing vaccination (MCV) recommendation by the World Health Organization (WHO) [16], however, with differential administration of the 2-dose approach where MCV1 is uniformly offered at 12 months, but varying timing of MCV2 across the ten provinces and three territories. Despite a strong institutional and organizational medical arrangement, immunisation is not mandatory at the national level in Canada. The provinces of Ontario, New Brunswick and Manitoba have implemented legislation that requires proof of immunisation for school entry [17] but immunisation, however, remains largely discretionary in other provinces. In many jurisdictions around the world, mandatory immunisation has been attempted for varying reasons with varying results, however, evidence did not support its effectiveness [18]. With the variations in the legal, policy and practice approach to immunisation, measles infection remains a source of concern in preventive medicine practice and research communities. Immunisation coverage rates measure the numbers of individuals who have received the appropriate doses by a specific date or age and are a reliable indicator of the preventative measures to control the spread of disease. Canada operates a goal of 95% coverage of one dose of measles immunisation by 2 years and 2 doses by 7 years [19]. The recent re-emergence of measles in some pockets of the population in Canada casts a doubt on possible elimination in the near term [20–22]. Some studies have linked the re-

emergence with a low level of herd immunity threshold (HIT) for the disease [23]. In particular, low population coverage with the recommended 2-dose regimen for measles vaccine by age of 2 years has been flagged as a causal factor [24].

Research into equity gaps in childhood immunisations exists but very little is known in a Saskatchewan context. Differences in immunisation coverage within and between health regions or geographical areas have been documented in the literature [25–27] with differences in health arising from social determinants of health between different groups making it challenging for some individuals and groups to integrate fully in the society, which affect such individual's health-seeking behaviour. Considering the established linkage between social determinants of health and health outcomes [28, 29], public health institutions are aware of the effect of health inequalities and inequities; however, evidence is not rife on whether these efforts have had impact on equity gaps. Several factors influence the rate of immunisation like poor access, low education, limited family support and poverty, religious beliefs and colony formation [30, 31] among others. Multiple chains of transmission have been documented among religious communities that actively oppose or resist immunisation efforts [11, 32–34].

The province of Saskatchewan routine measles immunisation schedule are two doses of measles containing vaccine with first dose recommended at one year (12 months) and a second dose starting from 18 months of age [35]. Publicly available data show an unequal distribution of measles immunisation coverage among the health regions in the province of Saskatchewan. In 2014, the province recorded 75.3% at 2 years MCV1 coverage and 91% for MCV2 at 7 years age [36], a range of 63.6% - 86.4% for MCV2 among regional health authorities in 2016 [37], and 79.9% average (69.1% - 93.2%) in 2018 [38]. Evidence from the Saskatoon Health Region (SHR) suggests that incomplete immunisation with the attendant low coverages is primarily associated with low income, single parenthood, cultural status, and differences in beliefs [39], and where immunisation disparities exist between rural and urban areas and from neighbourhood to neighbourhood.

Understanding the temporal trends in and drivers of measles immunisation coverage in small-area geographies of Saskatchewan can offer insight into how to improve immunisation policy development and implementation practices in the province of Saskatchewan and in contextually similar Canadian and international jurisdictions. Previous studies which looked at immunisation uptake in the province of Saskatchewan have examined the perspectives of the caregivers [27, 40, 41], but little is known on the perspectives of the health care providers as an important contributor to understanding coverage issues. Apart from the large regional health authorities such as Saskatoon and Regina Qu'Appelle Health Regions, which had carried out studies to document the coverage within their jurisdictions, there is scanty information on coverage rates at sub-regional levels, or within various quintiles of deprivation in the rest of the province. This study therefore examines measles immunisation coverage among health regions in the Canadian province of Saskatchewan and explores the barriers and enablers to achieving herd immunity threshold in the province and its smaller geographies.

## Materials and methods

### Study setting

The province of Saskatchewan is one of the thirteen provinces and territories of Canada. It is landlocked between the provinces of Manitoba and Alberta on the east and west respectively, Northwest Territories in the north and the United States of America to the south. It has an area of 651,900 km$^2$ and a population of 1.174 million inhabitants (2019). The province currently operates administratively through one health authority. Until the consolidation of all

regional health authorities (RHAs) into one Saskatchewan Health Authority in 2017, the health administration was carried out through Athabasca health authority located in the north of the province and twelve other health regions immediate to the south of Athabasca, namely Cypress, Five Hills, Heartland, Keewatin Yatthe, Kelsey trail, Mamawetan Churchill River, Prairie North, Prince Albert Parkland, Regina Qu'Appelle, Saskatoon, Sun Country, and Sunrise [42].

In this study, we define the concept of social factors for immunization as those related to the conditions in which people grow, live and learn including culture and social norms, also work and group conformity [43] while institutional factors were looked at in form of situations, policies or procedures that systematically disadvantage certain groups of people [44].

## Data source

The study adopts mixed methods of inquiry using historical measles immunisation quantitative data of 12–14 months and 18–24 months age groups and interviews of key informant in the province of Saskatchewan.

**Quantitative data.** A total sample of 16,582 children on first dose (MCV1) and second dose (MCV2) data from 2002–2013 were extracted from Saskatchewan Immunisation Management System (SIMS) registry. This study took place at a period when health system transformation was taking place in the province when they migrated from Saskatchewan Immunisation Management System Registry to the Panorama Gateway, and the only available data at the time of study were up to 2013. SIMS registry is an electronic secure storage of health records of public health services in Saskatchewan which includes immunisation, communicable diseases, and outbreak management accessible by authorized public health providers. We used annualized data from the two population groups aged 12–14 months and 18–24 months. We define immunisation coverage as the percentage of children who had their MCV1 between 12–14 months among the children population aged 12–14 months and percentage of children who had their MCV2 between 18–24 months among the children population aged 18–24 months. The two groups were chosen to conform with the Saskatchewan immunisation schedules [35].

**Qualitative data.** The qualitative part utilized interview data obtained from eighteen purposively selected participants interviewed at two different times. The interviewees included frontline officers and individuals with policy roles in their respective regional health authorities, such as public health nurses, immunisation administrators and medical health officers (MHOs). All the selected eighteen key informants participated in the first phase of interviews while seventeen participated in the second phase. The first stage of interviews which took place in the fourth quarter of 2016 collected the interviewees' initial general perspectives on factors that promote or hinder measles immunisation uptake in their health regions while the second stage, in the fourth quarter of 2017, was conducted to validate key themes and insights from the initial interviews and after quantitative results were presented to the participants. While the first interviews were by telephone, the second interviews were carried out in-person and both were audiotaped for in-depth analysis. Qualitative data triangulation [45] was ensured by interviewing two stakeholders each from each regional health authority. The inclusion of the qualitative strand was to provide context and possible explanations to the quantitative findings while embedding the study results with the experiences of the participants [46, 47]. The information asked from the participants pertained to their work in those periods when the quantitative data were collected; hence they were reflecting on their experience during the time data were collected, taking into account the past and future implications. Fig 1 displays the ten participating health regions key informants' selection criteria.

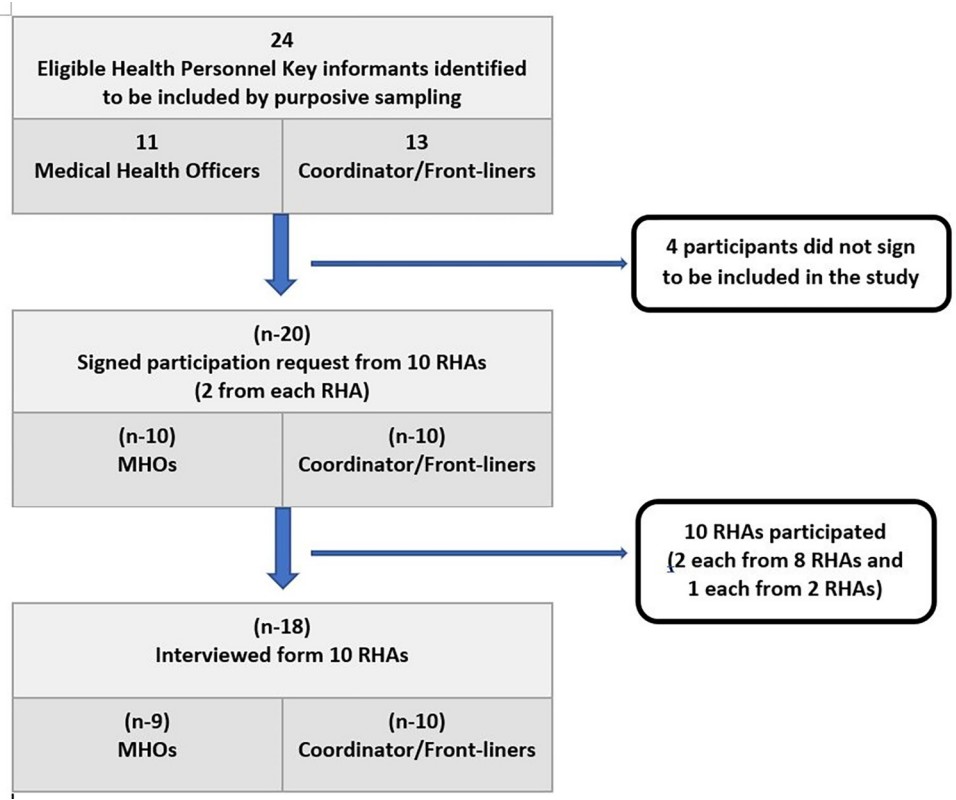

**Fig 1. Participants included in the qualitative strand of the study.**

## Ethical consideration

This study was approved by the research ethics board of the University of Saskatchewan, Canada (BIO-16-18), Regina Qu'Appelle Health Region Research Ethics Board (REB-16-91) and all the remaining 9 participating Regional Health Authorities of the province of Saskatchewan. The immunisation records data was extracted with researcher-generated syntax from the Saskatchewan Immunisation Database through Saskatchewan eHealth in a special format to ensure that the data remained anonymized and de-identified throughout the research process and collated through a provincial Public Health Observatory (PHO) leveraging upon the province data sharing agreement. All key informants had both written and verbal informed consents before interviews.

## Data analysis

**Quantitative.** The immunisation data was analysed using Joinpoint regression models methods. Joinpoint trend analysis regression software is useful to test whether an apparent change in trend over a time period is statistically significant or not [48]. Our analysis seeks to determine whether there is an increase or decrease over time in coverage rates and assesses the rate of change and the behavior of a response variable separately in characteristics of groups in the different periods of the explanatory variable [49, 50]. In this study, we evaluated the time trends in measles immunization coverage by geographical location and socio-economic deprivation quintile characteristics. We compared rates of changes for the measles immunization coverage by using the Annual Percent Change (APC) and Average Annual Percent Change

(AAPC) and the confidence intervals to determine an increase or decrease trend depending on whether the APC is positive or negative. When the lower confidence limit has a positive value, the rate of change over the time-period is depicted to be an increase and a decrease if the upper confidence limit is negative [50].

A ratio statistic was calculated to determine how the two groups relate to each other over the time period under observation, using R = (μ1 + 1) / (μ2 + 1), where APCs of the two groups are represented by μ1 and μ2 respectively [50]. The data generates outcome variables in percent coverage rate, and to use such data for Joinpoint regression, the percentages were converted to proportions, p. The standard error (SE) of the proportion was calculated using the formula below:

$$\text{Standard Error (SE) for large sample sizes} = \text{sqrt } [p \times (1-p)/n]$$

$$(\text{where p is the proportion, n the total population of the observed})$$

The percent coverage data calculated for all the birth years (2002–2013) from each health region were used and the response variable for the analysis of coverage was the natural logarithm of the percent coverage. The independent variable was the year from 2002 to 2013. The analysis fitted the simplest Joinpoint model that the data allowed with a statistically significant Joinpoint set at $p<0.05$. For graphical displays and visualizations, we used STATA$^{TM}$ 16 [51] to convert the Joinpoint results where necessary.

To assess socio-economic disparities, we used the index of deprivation developed at the Institut National de Sante Publique du Quebec (INSPQ) which measures deprivation at the level of dissemination areas (DAs), the smallest areas for which Census data are available in Canada, comprising of approximately 400 to 700 residents living in the same small area geography [52]. The deprivation index looked at social and material dimensions with the proportion of single parents, the proportion of residents living alone, and marital status in the social components, while the material deprivation measured educational attainment, average income, and employment status variables [52]. The DAs were divided into five quintiles 1 to 5 (Q1 to Q5) where each quintile represented 20% of population with Q1 being the most privileged and Q5, the least. We referred to these quintiles as Socio-economic deprivation quintiles (S-EDQ1 –S-EDQ5) in this paper.

**Qualitative.** The qualitative analysis employed hybrid inductive and deductive thematic analysis strategy described by Fereday & Muir-Cochrane [53]. This process included development of the codes, testing code reliability, summarizing the data and identifying the initial themes, code connection and theme identification and legitimization of the themes. The interview data were transcribed verbatim by a team of institution professionals. It was then read by the researcher thoroughly for immersive familiarization before coding. Nvivo version 12 [54] was used for the data analysis. Ideas in the interviews were first sorted into codes. The passages in the transcribed texts data were identified with respect to the concept being addressed by respondents and the relationship between the concepts followed by the categorization of the codes based on their similarities and relative differences and relationship to the research question and the comprehensive literature review. The categorized codes were then organized to subthemes and the themes informed the categories. We ensured that identified subthemes met recurrence and repetition criteria by ensuring a full understanding of the data to reduce biases and preconceived notions [55]. It was found that some sections or passages of some interview response fitted more than one theme and hence were coded in multiple ways depending on how many themes they fit into. The themes were refined after reviews by the research team as well as by presentation and discussing with the interview participants for legitimization of the coded themes and the categorization.

## Results

### Quantitative results

**Trends in measles immunisation.** We assessed the two-year age group for the study population demographics, there's a progressive increase of the study population from 2002 to 2009 reaching the highest figure of 15,189 in 2009 and decreased gradually to 14,106 by 2013 but without reaching the lowest figure of 13,273 of 2002 (Fig 2A). Two-thirds of Saskatchewan residents live in the cities as compared to one-third in the rural areas. There was also a progressive increase in the proportion of people living in the urban areas contrary to a gradual decline of population in the rural locations. Among the health regions, the RHAs with the three largest Saskatchewan cities, Saskatoon RHAs, Regina Qu'Appelle and Prince Albert Parkland saw a

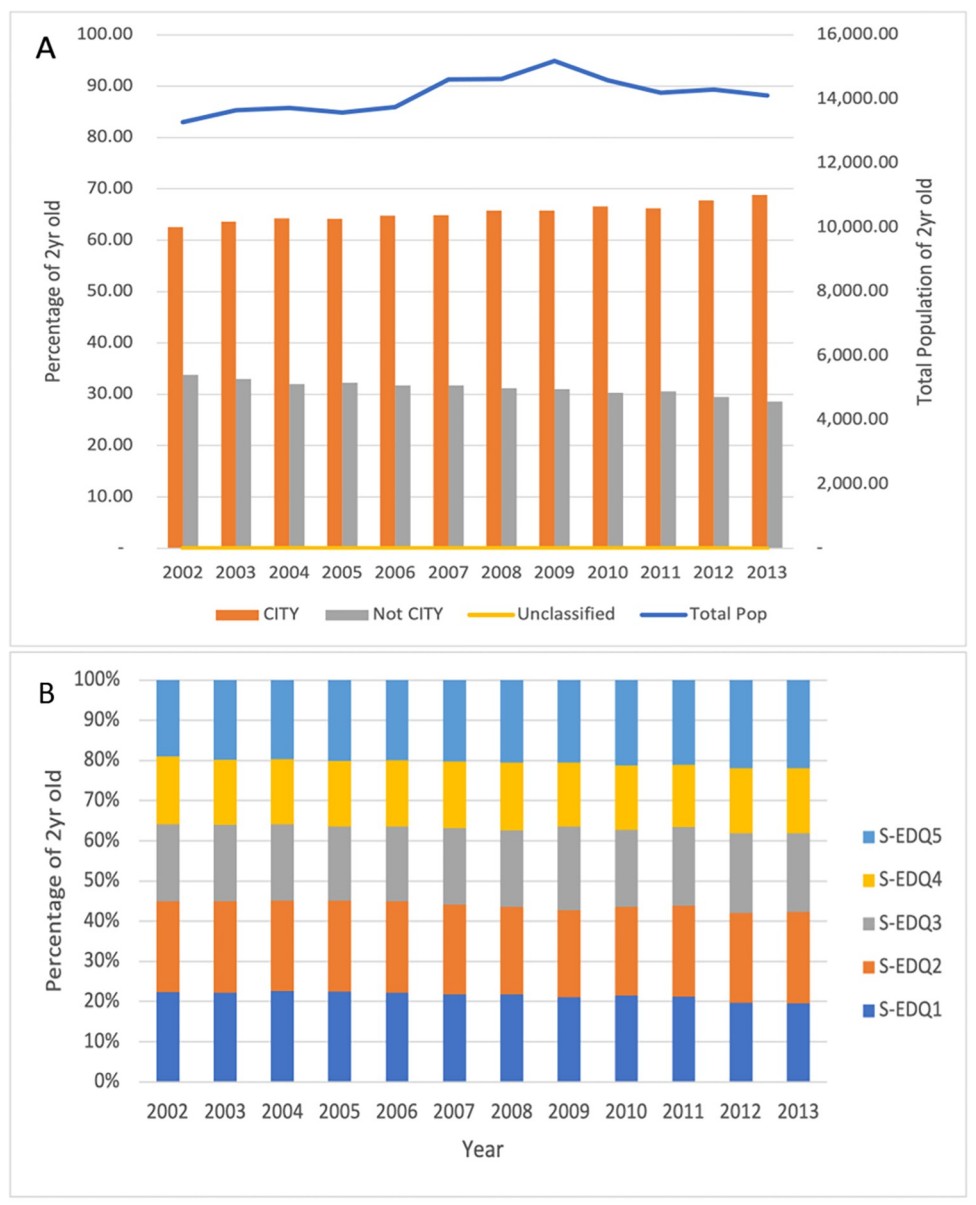

**Fig 2. Study sample characteristics.**

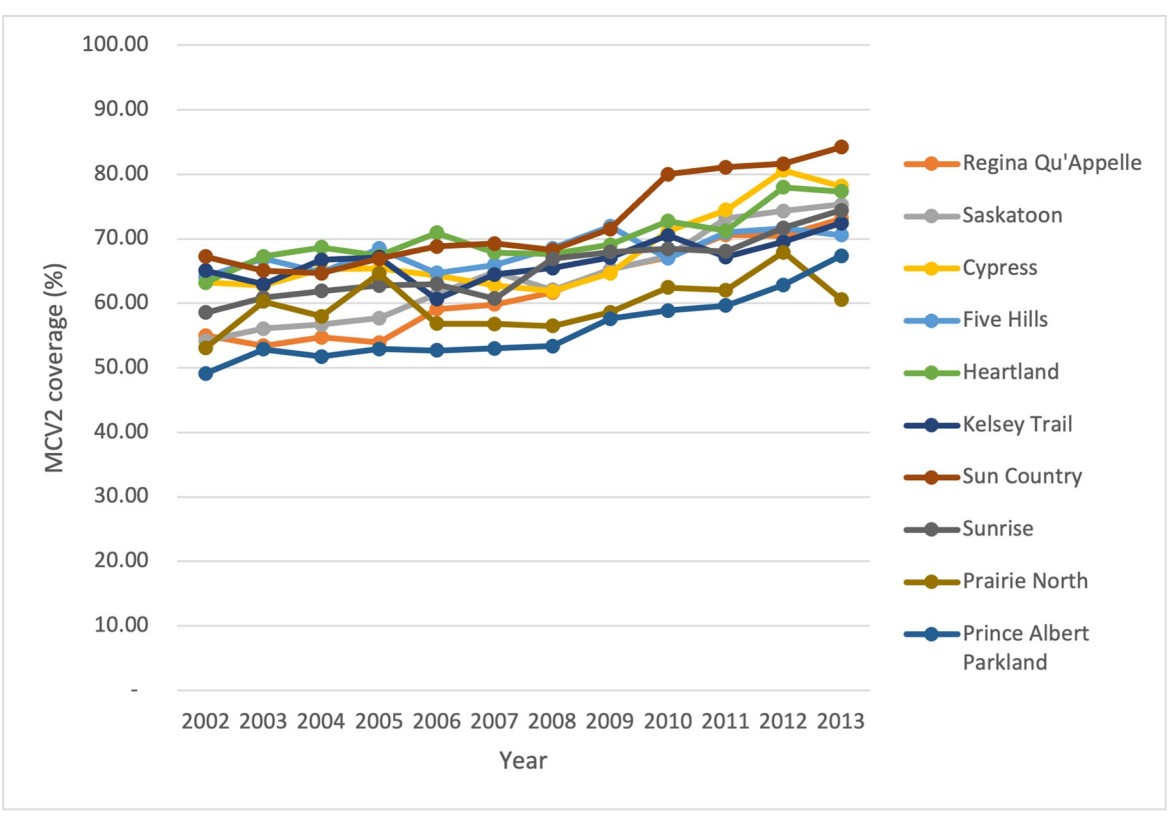

**Fig 3. Saskatchewan RHAs measles coverage among on-time for MCV2 at 24 months (2002–2013).**

progressive increase in population in the study period, with Saskatoon RHA having a higher level of increase than Regina Qu'Appelle and Prince Albert Parkland RHAs. The other health regions observed either a constant or a reduction in population between 2002 and 2013. The proportion of <2-year children and the socio-economic quintile they belong to range between 13.77% for S-EDQ4 in 2011 to 20.28% for S-EDQ2 in 2013 (Fig 2B). All the five (1–5) levels of socio-economic status are represented in the province.

From MCV2 results for 2002, Sun Country RHA had the highest coverage of 67.22% while Prince Albert Parkland RHA had the lowest at 49.15%. In 2013, Sun Country attained the highest coverage figure of 84.2% while Prairie North had the lowest coverage figure at 60.58% as compared with other RHAs. It should be noted, however, that there were varying coverage rates for each of the RHAs during the years 2002–2013. It is observed that Sun Country RHA reached a higher proportion of the target group than any other RHAs in the study period (Fig 3).

There was a progressive increase in MCV2 coverage in the province between 2002–2013 from 56.32% to 73.21% (16.89%-point increase) (Fig 4) The three segments represented by the APC were all significantly different from zero at 1.68, 4.09 and 2.14 for the periods 2002–2008, 2008–2011, and 2011–2013 respectively with an AAPC of 2.4 (95% C.I. 2.0–2.9, P < .005).

**Assessing disparity in measles coverage by geographical location.** There was increase in the coverage trends in Saskatchewan rural and urban locations with coverage rates being higher in the rural than in the urban locations (Fig 5). The coverage rate among on-time for second dose at 2-year (MCV2) increased between 2002 and 2008 (APC: 0.8, 95%CI: 0.1–1.6), followed by a higher progressive increase from 2008 till 2013 (APC: 3.0, 95%CI: 2.1–3.8) for

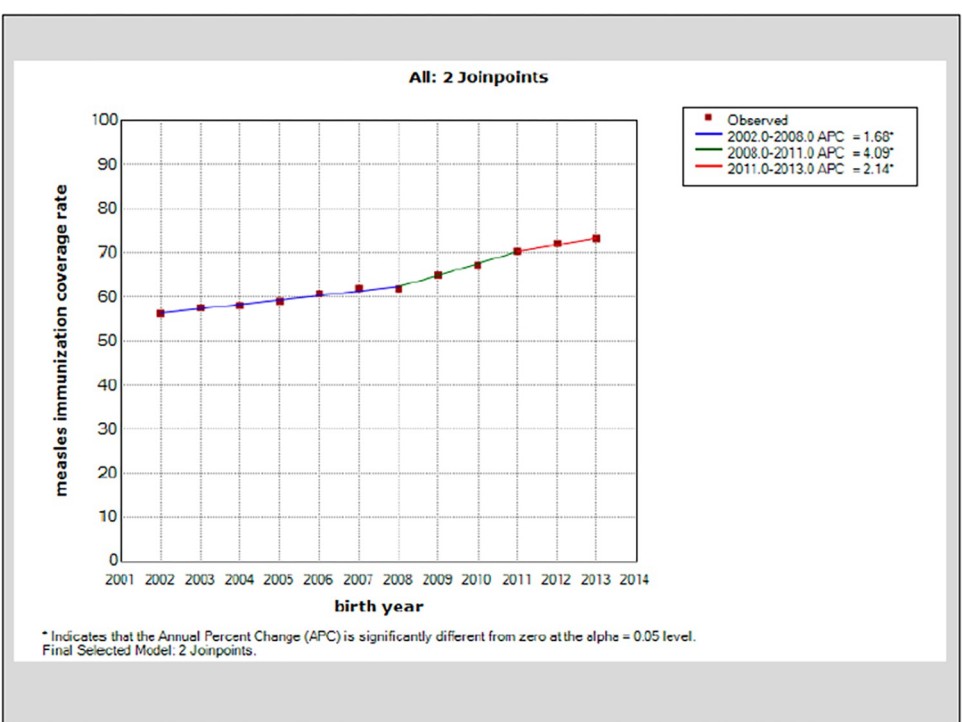

**Fig 4. Saskatchewan 10 RHAs average coverage for on-time for MCV2 at 24 months (2002–2013).**

the rural locations. The urban location coverage rates displayed an increase from 2002–2008 at an APC of 2.3, a further increase of APC of 4.5 from 2008–2011 with the rate of change being significant with 95%CIs of 1.7–3.0 and 1.6–7.5 respectively while the increase from 2011–2013 and an APC of 2.0, was not significant (95%CI -0.4–4.6). A test for parallelism of the two coverage rate trends for the on-time for MCV2 in the urban and rural locations to check for any difference between the two trend lines indicated no parallelism (p value = 0.027) indicating a significant difference between the trend lines of the two geographical locations.

**Assessing disparity in measles coverage by socio-economic status.** The relationship between the socio-economic status and measles coverage rates in the province was compared using the result of a Joinpoint regression which modeled coverage trends is shown in Fig 6. Although we have presented the five socio-economic deprivation quintiles to see the relationships, we compare S-E DQ1 and 5 Joinpoint fitted models in this research paper on the RHA pooled data for on-time for second dose at 2-year age (MCV2). There was increase in coverage rates from 2002–2013 (APC 2.0) in the S-EDQ1, an increase that was significantly different from zero (CI of 1.7–2.2). The S-EDQ5 coverage rates with a three-section Joinpoints displayed an increase from 2002–2008 with an APC of 2.8, a further increase to 5.8 from 2008–2011. In both of those APCs (2002–2008 and 2008–2011), the rate of change was significant with CIs of 1.9–3.7 and 1.6–10.1 respectively. The third Joinpoint for the S-EDQ5 showed a decrease from 2011–2013 with an APC of -0.2, which was not significant (CI -3.8–3.5). The two coverage rate trends for the on-time for second dose at 2-year age group in the S-EDQ1 and 5 were tested for parallelism (whether there was any difference between the two trend lines), and Parallelism was rejected (P < .001 at 95% CI).

Summarizing, there was progressive increase in the coverage rates for both S-EDQ 1 and 5 and among both rural and urban populations for MCV2 in the province of Saskatchewan. The

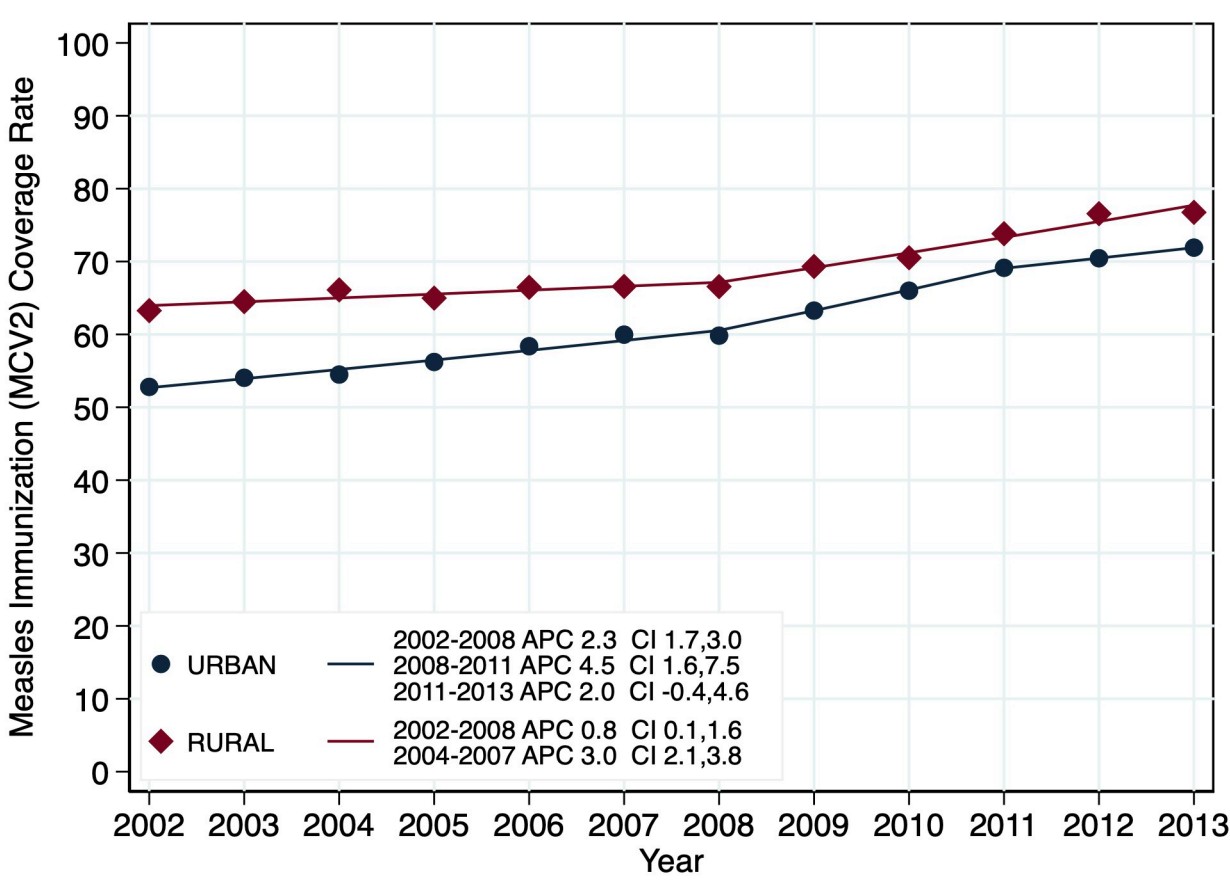

**Fig 5. Trends of MCV2 coverage rates in 10 Saskatchewan RHAs by geographical location (2002–2013).**

coverage rates were higher in the S-EDQ1 than S-EDQ5 and among the rural dwellers than the urban inhabitants among the on-time for MCV2 all through 2002–2013. Both S-EDQ and geographical location trend lines show convergence in coverage from 2002–2011 before a divergence between 2011–2013.

## Qualitative results

Eighty-three percent (n = 15) of the interview participants had at least eleven years' experience, either in their present positions or in Saskatchewan RHAs in various public and population health capacities; with MHOs having an average of thirteen years' experience and Coordinators/Front-line Immunisation Officers, an average of twenty years.

**Barriers to measles immunisation uptake.** The interview responses show that many health regions struggle to realize adequate measles immunisation coverage rates due to different barriers, including individual, region-specific, and cross-cutting institutional factors. These various barriers are described here.

*Access related issues*. Socio-economic factors impact the ability to bring the child to the clinic for immunisation due to lack of access to vehicles. Relatedly, it is reported that many new immigrants lack understanding of what health services are available and how to navigate the health system generally, and immunisation services specifically. Caregivers who are in the

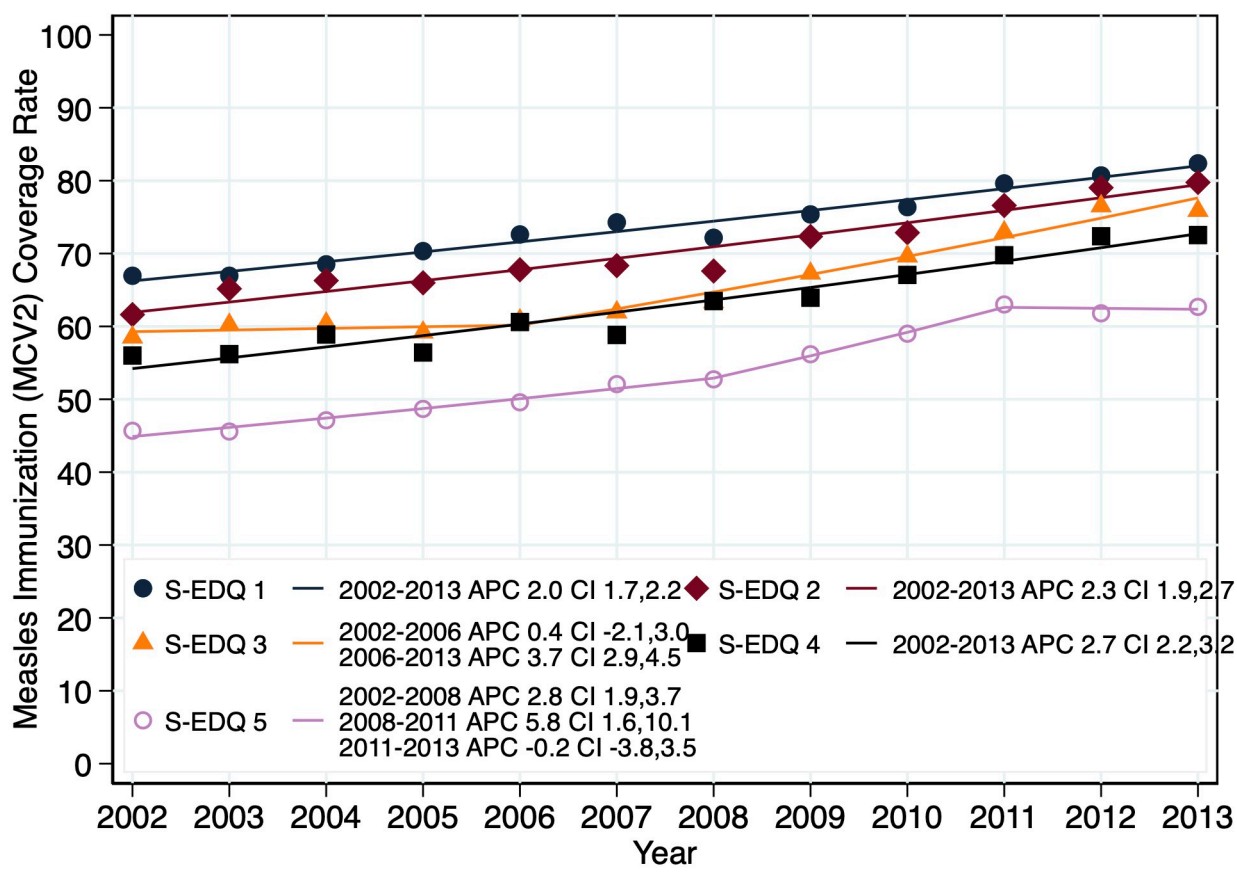

**Fig 6. Trend of MCV2 coverage rates in 10 Saskatchewan RHAs by S-EDQs (2002–2013): S-EDQ–Socio-economic deprivation quintiles ranges from 1 (affluent quintile) to 5 (poorest quintile).**

active workforce also have competing demands for their time, making it somewhat difficult to leave work to attend the clinic during work hours especially for those who are engaged in more than one job. *'Some clients have problems with location or time that the appointments are available to the general population, they are just not able to make it happen during the work hours'*. The interviewees also mentioned transience as a challenge to achieving higher coverage. Transient populations include the First Nations population who have a place of residence in the health regions but also live on-reserve and thus fall under the federal jurisdiction for health service provision. In Canada, First Nations people are often registered in the provincial system within the health regions data management platform as well as in the federal jurisdiction and are part of the denominator for coverage calculation. '*We do see some of our lower rates in rural areas that border with Federal Reserve. There is difficulty capturing immunisation coverage with children that may be going back and forth between a reserve and our health region'*. The systems level challenge to follow up with the immunisation status of this population is a major barrier to improving on the measles immunisation coverage rate in any of the health regions where this population exists because data are not shared routinely between federal and provincial jurisdictions.

*Fear and hesitancy issues*. Caregivers' decisions on vaccination are complex. Factors like experiences, information, even emotions and trust influence their decisions. The reluctance, delay in acceptance and sometimes refusal of vaccines despite vaccine services availability is

referred to as vaccine hesitancy [43, 56]. Personal factors such as fear, anxiety, and general skepticism of caregivers regarding the subject of immunisation was recurrent in the interviews, and all the participants responses centered on some caregivers not bringing their children for the required measles injections. One of the most mentioned is fear of needles according to one participant *"We definitely have some that just have fears and hesitancies of needles"* but also fears related to whether the measles vaccine causes autism or autism-related disorders, the possibility of reactions to the vaccine, distrust of the system, and other parental concerns. According to participants, some parents sometimes wondered '*how many is enough or too much*?' among the immunisation schedules that children have to take at every milestone in their early years, especially when measles vaccine is given in combination with other antigens in MMR and MMRV. Such hesitations can hamper the optimization of coverage rates where lack of education and engagement is missing at the grassroots level.

*Anti-vaccination issues.* Related to the parental fear factor mentioned above is the prevalence of anti-vaccination ideology among certain segments of the population. While fear and anxiety are dependent on individual characteristics and can be mitigated with public education and thorough engagement strategies, anti-vaccination ideologies take a philosophical objection to immunisation in general, and to multiple vaccines for infants specifically. According to some interviewees, who described some characteristics of the anti-vaxxers to the measles immunisation in Saskatchewan as follows '*. . .of course, there are anti-vaxxers. They don't have any immunizations. They don't go into the system at all where their name could be, and they're home-schoolers. So, they're not going to go in the school system. They're not in the government systems. . .*' Some participants also are sometimes influenced by unsubstantiated news in social media. Participants also mentioned the potential role of collective cultural beliefs where people in communal living arrangements or tightly organized religious, cultural, or social groups can exhibit "group-think" when anti-vaccination beliefs are promoted within the group. '*. . .we do have some religious group and then have people that just have philosophical objections for whatever reason, things they've read from the internet or the hype from people. . .*'. Participants identified several places in the province where local social or religious groups hold and promote anti-vaccine ideas where more targeted approaches to gaining trust may be required.

*Systemic and resource limitation.* The fourth set of barriers are related to institutional issues such as funding, staffing and media access. Interviewees are concerned about the shrinking funding of the health system in the province, which limits the amount of funds available to carry out public education at the community level, develop online platforms and materials to reach clients, and invest in transportation resources to reach remote settlements. '*. . .one of the main challenges is budget deficit and also overworked public health nurses.' 'I think another barrier we have is reaching people, the amount of resources we have makes it difficult to reach the "hard to reach people"*'. While systemic and resource limitations impact all health regions, they amplify existing challenges in small, often remote geographical locations that are far from large urban areas, particularly those serving predominantly Indigenous populations. There is also the issue of data access restriction between the federal and provincial health jurisdictions, and across provincial health regions. First Nations data in the federal system are not readily available on provincial immunisation registries [57] even though there is movement between the on-reserve and the off-reserve jurisdictions. '*. . .It's really not that the majority of kids are behind on immunisation it's just that there is incomplete records on panorama so it shows that they are behind but really if the reserve records were on the platform, those kids could be up to date. . .*'. As of the time of the study interviews, each region could only look at their own data, even though the Ministry of Health could look at the entire database for the province. Therefore, when people move from one region to another, there can be delays in updating records making it difficult to get an accurate picture of who needs immunisation in that region for timely

intervention. The integration of all health regions into 'one health authority' at the provincial level may have addressed this latter challenge, however, the lack of access to the First Nations data with the federal system remains an issue. Interviewees reported that such barrier leads to inaccuracy in immunisation coverage data, inability to identify specific demographics for targeted outreach, and conduct a thorough analysis of resources required to achieve herd immunity.

**Enablers to measles immunisation uptake.** Interview participants identified a variety of enablers to mitigate current challenges to measles immunisation uptake in the province. Some of the themes that emerged are in direct response to the barriers mentioned above—some of which are already in place in some health regions; however, they also highlight additional factors that could move the health authorities toward optimal measles immunisation uptake. These enablers are summarized below.

*Improved communication and increasing role of the media.* Almost all interviewees stress the importance of multiple channels of communication and engagement that include a reminder system, interprofessional communication and active community and stakeholder engagement. Reminder methods such as telephone, letters, emails, cards and the CANImmunize app (formerly immunizeCA app) can be helpful for clients to track and keep their immunisation schedule. For example, one interviewee suggests *". . .once the child was 13 months old and they hadn't received their 1-year immunisation, they'd get a little card in the mail that reminded them they needed to book an appointment."* Many health regions also promote the use of the CANImmunize app, an interactive app supported by the Public Health Agency of Canada to provide information about vaccines and immunisation frequently asked questions. Regular communication among health workers was also seen as an essential enabler for bridging knowledge gaps and to develop coordinated responses to client needs in order to build confidence in health care providers and the overall health system. In order to develop adequate knowledge, which is a prerequisite to organizational competency, training and workforce development require collaborations with other health care workers outside of the RHA, including physicians and Nurse Practitioners. At the individual level, some interviewees reported that the "handouts and pamphlets" provided at maternity wards upon delivery of babies have been helpful to engage new caregivers on immunisation conversations and to address any questions they may have. *'. . .We give handouts and discuss immunizations at the maternity visit program and home visit right after the baby is born.'*

While this is less frequent, some interviewees mentioned that the media has an important role to play in enabling measles and other forms of immunisation uptake in the province of Saskatchewan. *"We used it [the media] sometime for influenza and we had several people line up to get flu vaccine, even though we weren't appropriately staffed to manage the turnout, but we were happy to have the people. . . the media really does help"*. This includes the use of radio and TV to educate the public to encourage parents to immunize their children, and reiterates the importance of not only getting immunised but the timeliness of the immunisation: *"I think it would be great for the media to continue to supplement social media messages, but media does provide excellent service especially the radio stations,"* Despite the identification of a clear role for traditional media such as radio and TV, only a few interviewees mentioned the important role the social media can play and it is in regard to the fact that *"we have not explored the use of social media to their full potentials"*.

*Community engagement.* At the community level, community awareness outreach helps to stimulate conversation about measles immunisation especially in communities where they were beginning to see low attendance and uptake of immunisation services generally: *"We have gone to a couple of health fairs or promotional days where we actually take immunisation displays with us and just talk about not just measles but other vaccines"*. Interviewees also mentioned the need to coordinate communication with school districts through school nurses

especially when the children have missed the opportunity to take the measles vaccines at the appropriate periods. For example, "*letters are sent home to the caregiver for consent to administer the vaccines that were due to older children*" These school reminders are helping to improve coverage for non-school age kids that are overdue for immunisation as they serve as triggers for caregivers about other children in their homes that are behind on routine immunisation.

*Flexibility in service delivery.* Another enabler identified by many interviewees is the need for flexibility in service delivery to accommodate different socioeconomic groups. These include flexible timing for services and creation of more drop-in clinics and sites for Low Socioeconomic (LSE) groups, some of whom may not own phones and hence telephone reminder messages may not reach them. Information tracking and evaluation is also linked to the perception of flexibility in service delivery. Many interviewees suggest running through overdue lists regularly, creating regular status reports of coverage to identify where to direct resources for maximum results. The number of sites, coverage status reports and client appointments are regularly reviewed and if necessary revised to support LSE clients. Through interactions with clients, health care workers also assess client attitudes that point to vaccine hesitancy so that the challenge can be addressed pro-actively: "*We are analyzing which areas of the region have low coverage like we have done in the rural areas as well, as I mentioned before, the hesitancy group which are in . . .. So, these are some cases where we are trying to identify whether it has to do with the access or might be some hesitant group that we want to go after.*"

*Targeted social responses.* In addition to communication, engagement and flexibility, some interviewees mentioned the necessity to address clients' other needs, including home visitation, provision of transportation or bus passes, rewards in the form of gift cards, where and when possible. According to aggregate interview responses, three health regions are already providing these as enablers to increase access, although only two of the three health regions explicitly linked the practice to improved measles immunisation. In addition, these specific interviewees also indicated that different incentives work for different groups of people and suggested the need for each health region to understand their client characteristics before introducing any incentive. For example, "*incentives work actually better in lower socio-economic neighborhoods*" and with "*our inner-city population.*" "*We would buy books for kids, and we have gifts cards from Safeway or Co-op that we would give to the parents*".

## Discussion

In this paper, we present the measles immunisation coverage in the province of Saskatchewan focusing on the trends, barriers, and enablers to identify where actions are required for improved uptake which could move the province towards achievement of a herd immunity. The study used both quantitative and qualitative data to explore trends, patterns, and issues in the province and by extension to similar health jurisdictions in Canada and internationally. Fig 7 summarizes the results with various issues and clusters representing key building blocks —individual, social, and institutional—that should be considered towards optimizing immunisation in the province. Each of these issues has critical implications for equity and public health policy. The framework includes individual, social and institutional dimensions of existing barriers and enablers and suggests a holistic, interconnected approach that integrates issues such as timeliness of measles immunisation dosages, population growth, urban-rural dichotomy, and regional differences and disparities in coverage rates.

### Data trends and regional disparities

The practice of immunisation trends monitoring is primarily reliant on coverage rates for its estimation and utilizes data which are routinely collected as a proportion of prescribed target

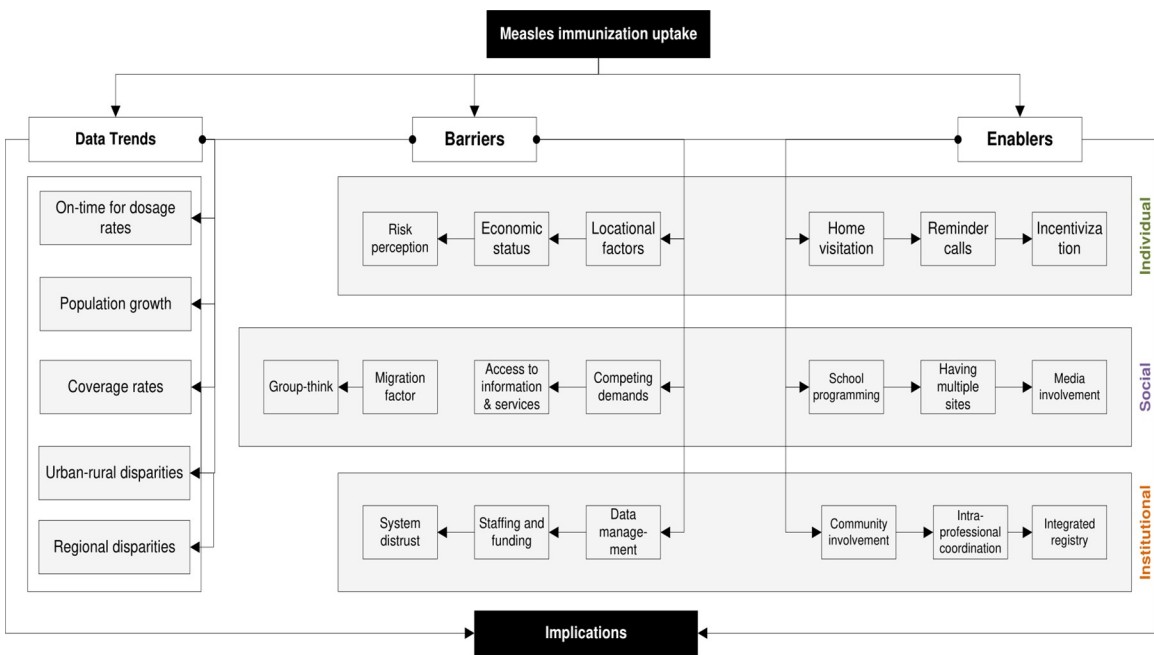

**Fig 7. Policy and research interventions for measles immunisation uptake.**

populations [58–61]. It is information from such coverage rates that are needed for required vaccine-needs projections and immunisation program planning and implementation, including the evaluation of progress toward both national and international goals [62]. The coverage goal for childhood vaccines developed in 2017 was 95% at 24 months for all publicly funded vaccines in all Canadian provinces and territories [63]. This goal is consistent with the WHO, and 2012 Global Vaccine Action Plans which provided a framework for more equitable access to vaccines currently in use for the full benefits derivable from them [62, 64], while reflecting the Canadian context which has been adopted since 1996. The trend in measles immunisation coverage across the ten health regions increased steadily from 2002–2013. The increase in trend may be a response to all the various health region-specific innovations and program implementation practices during the study period. While this is a positive sign, the HIT has not yet been reached, and in order to reach it, regular reporting on coverage rate at small area level with targeted approaches to under-immunized sub-populations will be required.

### Contextual adaption of interventions (incentives, targeted response)

Contextual adaptations were mentioned by interview participants as an enabler of immunisation coverage. Particularly, regional health authorities where targeted approaches are introduced observed improvement in immunisation uptake by the initiation and maintenance of community program builders recruited from the community to reach out to the parents and clients in the inner-city core neighborhoods in Saskatoon RHA in 2007. These targeted approaches are valued by individuals and communities, not only for measles immunisation effort but also for opportunities to address clients' other needs such as the determinants of the clients' overall well-being [65]. This finding suggests that interventions and implementation practices do not have to take the form of 'one size fits all'; rather, the introduction of culturally relevant and contextually appropriate approaches may be more effective than a standardized, universal approach to client engagement and outreach. This approach is similar to what was

described as proportionate universalism by Carey et al, 2015 [66] and Egan et al, 2016 [65]. The above efforts are part of the strategies and interventions that increased uptake of measles immunisation especially by bridging the gap in the socio-economic divide [65, 67] with a resultant improvement in the coverage rates over the period under study. These innovations and practices were confirmed by the key informant qualitative findings of this study. During the targeted approach period starting from 2007, many regional health authorities in the province of Saskatchewan utilized reminder systems more effectively, increased the number of immunisation service delivery points and became more flexible in terms of clinic opening hours to allow access for caregivers who could not attend within the strict regular opening hours. The reminder call systems have been used in other similar situations with improved coverage results [68–70]. All these efforts are towards tailoring immunisation service to the targeted beneficiaries [71]. These approaches were employed in different combinations by RHAs in the past, currently and into the future with modifications depending on contexts [70].

## Cross-regional knowledge-sharing, community engagement, and collaborations

Prior to the implementation of "one-health authority" in the province of Saskatchewan, each of the 13 health regions operated a more siloed approach in terms of operational framework and data management. Community engagement and outreaches were tailored to each region's context and managed by health professionals assigned to manage immunisation programs. While some of the gaps associated with this approach may have been bridged, local adaptation of innovations to improve coverage rates according to local need is desirable but learning from what has worked in similar situations is best in speeding up the "diffusion of innovation". Both quantitative and qualitative data show some regions have better coverage rates due to geographical advantages and innovative practices. For example, while Regina and Saskatoon RHAs saw a progressive increase in coverage rates throughout the period under study, Sun Country attained the highest coverage figure of 84.2% for the on-time for 2nd dose measles vaccine. Flexibility in service delivery, a needs-based approach that considered client socio-economic position, incentivization are some of the areas where cross-regional knowledge sharing will be valuable to practice. Proactive community engagement has been found effective at improving immunisation uptake by reducing negative group-think and decisions that lead to community resistance [72, 73]. It should also be noted that proactive engagement with clients and communities help health care workers to assess client attitudes that point to vaccine hesitancy with an opportunity for such to be addressed thereby reducing resistance [72]. While community engagement has been found to be a great tool to build relationships and trust to reduce vaccine hesitancy, this approach however, requires continuous collaboration among health professionals, local communities' stakeholders, and decision-makers in planning, designing, and implementing initiatives that support such practices.

## Prevalent institutional and policy issues

The study also found that institutional factors such as funding, staffing and database integration remain a barrier for measles immunisation uptake. This was particularly the case for health regions covering rural, often-remote locations, which have transportation access challenges. In some cases, confusion about whether immunisation records of individuals in Indigenous communities are up-to-date due to lack of coordination and linkage between the federal and provincial immunisation registries exist. Increased funding could help bridge staffing and resource issues, and more easily encourage health care providers and individuals with policy

roles to explore options for better coverage. The registry coordination challenge, however, is not new and has been raised in other similar studies, especially with respect to the lack of access to First Nations immunisation data to calculate provincial coverage rates [57, 74]. Moreover, due to the mobility of some First Nations people between on-reserve and off-reserve homes, tracking and communicating immunisation updates may be duplicated or sometimes omitted. It is therefore imperative that a national immunisation registry is created to facilitate greater transparency, coordination and consistency in tracking coverage rates in provinces and to develop a more accurate, timely and meaningful national outlook for measles immunisation coverage [57].

### Culturally responsive and context-specific vaccination messaging

Social and community beliefs can discourage individuals from accessing measles immunisation [75, 76], and health care providers do not have the authority and resources to address this challenge since immunisations are recommended and not mandated public health policy in many jurisdictions [77, 78]. The scale of hesitancy and resistance is more pronounced in some religious communities [79] but are also amplified by access to social media where 'pseudo-science' circulate more easily [80, 81]. Several factors, including conspiracy theories, general distrust, belief in alternatives, and concerns about safety have been identified as the drivers of the resurgence of vaccine hesitancy in recent past [43, 80, 82]. Hospital admission for measles can be as high as 47% among unimmunised individuals who were age-eligible for measles immunisation and up to 53% may develop acute respiratory failure [83]. Health care providers can only do so much as the legal environment allows, which suggests that persuasion and education remain the only tools to engage 'anti-vaxxers'. Therefore, there is need for a more coordinated approach within the health community and across government social sectors to engage individuals and communities where objection to measles immunisation is still prevalent to increase the profile of credible sources of vaccine safety information. Interview responses have established the benefits of the traditional media (TV and Radio) as a major enabler, but gaps still exist on the use of social media [84]. The need for staff and client social media literacy skills and education therefore becomes more imperative and urgent to enable the public to decipher credible sources of information on measles immunisation.

### Conclusion and public health implications

This study found that despite individual, social and institutional barriers to measles immunisation in the province of Saskatchewan, there has been a significant increase in uptake in all regional health authorities over the study period (2002–2013). However, the uptake is occurring at different average annual percent change in each region but with overall aggregate increase at the provincial level. While access related issues, fear and hesitancy of caregivers, anti-vaccination ideologies, and myriads of systemic and resource limitations were identified as key barriers in the province, enablers such as improved communication and community engagement, flexibility in service delivery, targeted social responses, and increasing role of the media are helping to address these issues. This suggests that there are opportunities for improvement, especially in the aspects of knowledge exchange, public engagement, and contextual adaptations of innovative or proven practices in a way that is sensitive to individual regional experience with measles immunisation. In the province of Saskatchewan, achieving HIT which is the desirable outcome of measles immunisation has not yet been reached, and for this to happen, regular reporting on coverage rate at small area level with targeted approaches to under-immunized sub-populations will be required.

While this research furthers existing knowledge on measles herd immunity threshold and can ultimately inform policy directions on measles immunisation, the results should be considered in light of its limitations. Firstly, only 10 of the 13 RHAs participated in the study. While these 10 RHAs represent 96.5% of the study population, the specific context, and characteristics of the non-participating RHAs may have been missed. Secondly, the Saskatchewan Immunisation Management System (SIMS) data used for the quantitative strand of the study was between 2002 and 2013—which could be considered relatively dated, although this range represented the entire dataset that was available in that system before Saskatchewan transitioned to the Panorama Gateway in 2014. However, the 2016 and 2017 qualitative data provided by key informants, majority of who were long serving in the RHAs were valuable in providing a comprehensive picture of trends and challenges faced in their respective RHAs addressing both interventions in the retrospective periods dating to a decade and how the new information will be used to make changes in the future. An apparent limitation of interview data analysis is information recall bias. It should also be noted that data were not available for ethnicity and immigration status. Future studies are needed to explore patterns of measles immunisation among other sub-populations which frequently exhibit health inequities such as immigrants and indigenous population. Overall, measles immunisation remains an important public health issue, therefore, studying the trends and patterns in measles immunisation uptake and the associated barriers and enablers in a typical Canadian jurisdiction such as Saskatchewan is an important issue for public health discourse, policy, and research internationally.

## Supporting information

**S1 File. Interviews guide for this study.**
(DOCX)

**S1 Fig. Saskatchewan geographical distribution by proportion of <2year children (2002–2013).**
(DOCX)

## Acknowledgments

MI would like to thank the public health nurses, immunisation administrators and medical health officers who were the interview respondents and also the University of Saskatchewan, Saskatoon, Saskatchewan, Canada for his doctoral support through the College of Medicine Graduate Award (CoMGRAD), National Science and Engineering Research Council (Integrated Training Program in Infectious Disease, Food Safety and Public Policy (ITraP) and Government of Saskatchewan Innovation and Opportunities Scholarship (SIOS). The content and opinions expressed in this manuscript is solely those of the authors and does not necessarily reflect the official views of the supporters or their affiliates.

## Author Contributions

**Conceptualization:** Marcus M. Ilesanmi, Cordell Neudorf.

**Data curation:** Marcus M. Ilesanmi.

**Formal analysis:** Marcus M. Ilesanmi.

**Investigation:** Marcus M. Ilesanmi, Cordell Neudorf.

**Methodology:** Marcus M. Ilesanmi, Sylvia Abonyi, Punam Pahwa, Volker Gerdts, Cordell Neudorf.

**Project administration:** Cordell Neudorf.

**Resources:** Sylvia Abonyi, Punam Pahwa, Cordell Neudorf.

**Software:** Marcus M. Ilesanmi, Cordell Neudorf.

**Supervision:** Sylvia Abonyi, Punam Pahwa, Volker Gerdts, Cordell Neudorf.

**Validation:** Marcus M. Ilesanmi, Volker Gerdts.

**Visualization:** Marcus M. Ilesanmi, Sylvia Abonyi, Punam Pahwa, Michael Scwandt, Cordell Neudorf.

**Writing – original draft:** Marcus M. Ilesanmi, Sylvia Abonyi.

**Writing – review & editing:** Marcus M. Ilesanmi, Sylvia Abonyi, Punam Pahwa, Volker Gerdts, Michael Scwandt, Cordell Neudorf.

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
