## [Decision Letter · Decision Letter 0]

4 Dec 2021

PONE-D-21-15652Trends, Barriers and Enablers to Measles Immunization Coverage in Saskatchewan, Canada: a Mixed Methods StudyPLOS ONE

Dear Dr. Ilesanmi,

Thank you for submitting your manuscript to PLOS ONE. After careful consideration, we feel that it has merit but does not fully meet PLOS ONE’s publication criteria as it currently stands. Therefore, we invite you to submit a revised version of the manuscript that addresses the points raised during the review process.

ACADEMIC EDITOR:  

The academic editor served as the second reviewer nd agreed major revision need to be addressed in order for the manuscript to be considered for publication.

We look forward to receiving your revised manuscript.

Kind regards,

Joseph Telfair, DrPH, MSW, MPH

Academic Editor

PLOS ONE

Journal Requirements:

Additional Editor Comments (if provided):

The academic editor served as the second reviewer nd agreed major revision need to be addressed in order for the manuscript to be considered for publication.

Reviewers' comments:

Reviewer's Responses to Questions

**Comments to the Author**

1. Is the manuscript technically sound, and do the data support the conclusions?

Reviewer #1: Partly

2. Has the statistical analysis been performed appropriately and rigorously? 

Reviewer #1: Yes

3. Have the authors made all data underlying the findings in their manuscript fully available?

Reviewer #1: Yes

4. Is the manuscript presented in an intelligible fashion and written in standard English?

Reviewer #1: Yes

5. Review Comments to the Author

Reviewer #1: General comment

The topic of the article proposed by the researchers is interesting, it is a contribution to the analysis of measles vaccination trends, from a mixed approach. The data is limited to the province of Sasketchewan, but could be understood given the qualitative approach the authors add to the study.

However, I am concerned about the lack of concurrency in the temporality of the quantitative and qualitative approaches. I can understand that the quantitative data for this study only includes up to the year 2013, while the qualitative analysis would correspond to a more current information collection (the authors do not report on this). If so, this temporal gap would mean that the social, historical and cultural context in which the analysis is carried out is not the same. Ideally, I would suggest to the authors to extend the trend analysis to a more current date, closer to the period in which the data from the qualitative approach were collected.

Here are some questions and suggestions for authors.

Summary

Line 32. It is not clear in what time or periods of time the interviews with the key informants were taken.

Introduction

Lines 75-78. Specify in the province of Saskatchewan, what is the regular schedule of measles vaccination in the first and second doses.

Materials and methods

Line 145. Specify if the vaccination coverage was obtained: annually? For the entire province or for each of the regions?

Justify why a data analysis is performed only up to 2013, why more recent data is not included.

Lines 153-156. Describe in what time period the key informant interviews were conducted. Describe whether there was triangulation of qualitative information and how it was done.

How do the authors justify a quantitative analysis of data from 2010-2013, while the qualitative analysis corresponds to a current period of time? That is, although there is a mixed approach to the research, the authors have worked with a quantitative and qualitative approach at two different, non-concurrent moments.

Statistical analysis

An analysis of differences by socioeconomic quintiles is carried out in the results. It is necessary to describe in the Statistical analysis section where this information was obtained and how it was included in the analysis.

I consider that, when evaluating the disparity in measles coverage by socioeconomic status, the authors could delve deeper into the results, there are some inequality analyzes that could perhaps be applied in the present study, as well as exploring differences by urban, rural, between migrants, between indigenous and non-indigenous populations

Results

Lines 223-224. Are you referring to first or second dose coverage for measles, or full schedule coverage?

Line 240-251. Specify what S-EDQ1 means. It is the first time that the abbreviation has been used but I cannot find what it corresponds to in the text. In addition to mentioning it in the Abbreviations section, it should also be included in the main text.

Again, I would suggest including additional analysis on inequality in vaccination coverage, for example, differences by urban and rural sector, between migrants, between indigenous and non-indigenous populations. Particularly, in the qualitative part, the differences between the urban and rural sectors are mentioned; It would be interesting to see if these differences are evident in the quantitative part.

Discussion

Line 430. What does HIT mean?

Line 433. To what period of time do these contextual adaptations refer (between 2002-2013? Or later). Since when did promoters of community programs begin to be introduced? Could it be observed if the presence of community program promoters really had an effect on vaccination coverage?

Line 448. What period of time do you refer to?

Line 454 and following. I suggest, for the reader's clarity, to make a summary table of the changes over time in health strategies and policies related to vaccination, which are mentioned both in the results and in the discussion.

Figures.

Figure 2. Include a note at the bottom of the figure that explains the graph and legends. Explain what S-EDQ is.

Figure 5. Include a footnote to the figure that explains the graph and captions. Explain what is DEP 1, 2 etc.

Figure 6. Include title, explain the main elements of the figure in a footnote.

6. PLOS authors have the option to publish the peer review history of their article (what does this mean?). If published, this will include your full peer review and any attached files.

Reviewer #1: No

---

## [Author Response · Author response to Decision Letter 0]

16 Feb 2022

Authors’ response: Trends, Barriers and Enablers to Measles Immunization Coverage in Saskatchewan, Canada: a Mixed Methods Study

Reviewer #1: General comment

R1.1: The topic of the article proposed by the researchers is interesting, it is a contribution to the analysis of measles vaccination trends, from a mixed approach. The data is limited to the province of Saskatchewan but could be understood given the qualitative approach the authors add to the study.

Authors response: Thank you for your constructive comments

Changes made: None

R1.2: However, I am concerned about the lack of concurrency in the temporality of the quantitative and qualitative approaches. I can understand that the quantitative data for this study only includes up to the year 2013, while the qualitative analysis would correspond to a more current information collection (the authors do not report on this). If so, this temporal gap would mean that the social, historical and cultural context in which the analysis is carried out is not the same. Ideally, I would suggest to the authors to extend the trend analysis to a more current date, closer to the period in which the data from the qualitative approach were collected.

Authors response: We looked more into contextuality in this study. The study looked at data from 2002 to 2013 which were the available data in the Saskatchewan Immunization registry at the time of the study. We intentionally used data to make the key informants (mostly experienced managers) aware of the gap and or challenges in measles immunization coverage to have them pursue vigorously quality and equity initiatives. The first interview with the key informants took place in the third quarter of 2016 at which time the only available data on the SIMS registry were updated until 2013 and the interviews were with people who had been working in the system for many years and they reflected over that history and brought to fore what they have done in that period. The information pertaining to their work in those periods when the data were collected were asked. The information acquired were aimed at informing future interventions and practice. We conducted the first interview when they had not seen the findings from the quantitative analysis. The coverage rates that the RHAs had seen before our study was the overall coverage for their respective RHAs as made available by the Ministry of Health in a non-granular form like our study did. Our study looked at rural-urban disparity in coverages as well as different socio-economic groups. We then collated the generalized themes to see what were done in different RHAs and compared that with the outcome in terms of coverage. Subsequently, we shared the results of the quantitative analysis with the same group of interviewees to see if they have made improvements or not especially with new evidence and the new details on geographical and socio-economic analysis. With that level of analysis, we carried out knowledge translation with the informants. This led to the second stage interviews in 2018, when we interrogated the same set of people about the past and reflect on how they wanted to use the new information to make future programmatic changes. We think their responses reflected not just that point in time but their experience over the last decade and their future plans, therefore, there is no substantial gap that could have affected the results. 

Changes made: Page 7, line 168 – 174: Eighty-three percent (n=15) of the interview participants had at least eleven years’ experience, either in their present positions or in Saskatchewan RHAs in various public and population health capacities; with MHOs, an average of thirteen years’ experience and Coordinators/Front-line Immunization Officers, an average of twenty years. The information asked from the participants pertained to their work in those periods when the quantitative data were collected; hence they were reflecting on their experience during the time data were collected, taking into account the past and future implications.

Also, page 23, lines 563 – 567: However, the 2016 and 2017 qualitative data provided by key informants, majority of who were long serving in the RHAs were valuable in providing a comprehensive picture of trends and challenges faced in their respective RHAs addressing both interventions in the retrospective periods dating to a decade and how the new information will be used to make changes in the future.

Summary

R1.3: Line 32. It is not clear in what time or periods of time the interviews with the key informants were taken.

Authors response: We have added the time of the interview

Changes made: Page 2, Lines 30 – 33: We utilized the entire population of 16,582 children under two years of age available in the Saskatchewan Immunization Management System (SIMS) registry for 2002 and 2013 in aggregate format and interviewed 18 key informants in pre-determined two-stages in 2016 and 2017.

Introduction

R1.4: Lines 75-78. Specify in the province of Saskatchewan, what is the regular schedule of measles vaccination in the first and second doses.

Authors’ response: The routine schedule for first and second measles vaccination has been specified.

Change made: Page 3, lines 73 – 75: The Saskatchewan routine measles immunisation schedule are two doses of measles containing vaccine with first dose recommended at one year (12 months) and a second dose starting from 18 months of age.[1]

Materials and methods

R1.5: Line 145. Specify if the vaccination coverage was obtained: annually? For the entire province or for each of the regions?

Authors’ response: As recommended, we have added “annualized”

Changes made: Page 6, line 151 – 152: We used annualized data from two population groups, aged 12-14 months and 18-24 months.

R1.6: Justify why a data analysis is performed only up to 2013, why more recent data is not included.

Authors’ response: This study took place at a period when health system transformation was taking place hence, we used all the available data up to that point.

Change made: Page 6, lines 145 – 149: This study took place at a period when health system transformation was taking place (i.e., from Saskatchewan Immunization Registry to the Panorama Gateway). The only available data at the time of study were up to 2013, hence the study used the total sample available in the province of Saskatchewan Immunization Management Registry.

R1.7: Lines 153-156. Describe in what time period the key informant interviews were conducted. 

Authors’ response: The time period has been specified. 

Change made: Page 7, lines 159 – 164: All the selected eighteen key informants participated in the first phase of interviews while seventeen participated in the second phase. The first stage of interviews which took place in the fourth quarter of 2016 collected the interviewees’ initial general perspectives on factors that promote or hinder measles immunisation uptake in their health regions while the second stage, in the fourth quarter of 2017, was conducted to validate key themes and insights from the initial interviews and after quantitative results were presented to the participants 

R1.8: Describe whether there was triangulation of qualitative information and how it was done.

Authors’ response: To ensure consistency in information received from each of the health regions, we interviewed two key informants.

Change made: Page 7, line 165 – 166: Qualitative data triangulation was ensured by interviewing two stakeholders each from each regional health authority.

R1.9: How do the authors justify a quantitative analysis of data from 2010-2013, while the qualitative analysis corresponds to a current period of time? That is, although there is a mixed approach to the research, the authors have worked with a quantitative and qualitative approach at two different, non-concurrent moments.

Authors’ response: See response above (R1.2)

Change made: Response R1.2 above

Statistical analysis

R1.10: An analysis of differences by socioeconomic quintiles is carried out in the results. It is necessary to describe in the Statistical analysis section where this information was obtained and how it was included in the analysis.

Authors’ response: We have described how socioeconomic quintiles were estimated in the methods section

Changes made: Page 9, lines 212 – 221: To assess socio-economic disparities, we used the index of deprivation developed at the Institut National de Sante Publique du Quebec (INSPQ) which measures deprivation at the level of dissemination areas (DAs), the smallest areas for which Census data are available in Canada, comprising of approximately 400 to 700 residents living in the same small area geography[50]. The deprivation index looked at social and material dimensions with the proportion of single parents, the proportion of residents living alone, and marital status in the social components while the material deprivation measured educational attainment, average income, and employment status variables.[50] The DAs were divided into five quintiles 1 to 5 (Q1 to Q5) where each quintile represented 20% of population with Q1 being the most privileged and Q5, the least. We referred to these quintiles as Socio-economic deprivation quintiles (S-EDQ1 – S-EDQ5) in this paper.

R1.11: I consider that, when evaluating the disparity in measles coverage by socioeconomic status, the authors could delve deeper into the results, there are some inequalities analyzes that could perhaps be applied in the present study, as well as exploring differences by urban, rural, between migrants, between indigenous and non-indigenous populations

Authors’ response: Your suggestions are well noted. We have included information on urban and rural analysis in the results section as well as in the discussion. However, immunization data by ethnicity and immigration status were not systematically collected in the province of Saskatchewan. Also, the authors did not have access to individual but aggregated data for the study. We have hence noted this as part of study limitations.

Changes made: Pages 11 – 12, lines 268 – 278: 

Assessing disparity in measles coverage by geographical location

There was increase in the coverage trends in Saskatchewan rural and urban locations with coverage rates being higher in the rural than in the urban locations. The coverage rate among on-time for second dose at 2-year (MCV2) increased between 2002 and 2008 (APC: 0.8, 95%CI: ), followed by a higher progressive increase from 2008 till 2013 (APC: 3.0, 95%CI: ) for the rural locations. . The urban location coverage rates displayed an increase from 2002 – 2008 at an APC of 2.3, a further increase of APC of 4.5 from 2008 – 2011 with the rate of change being significant with CIs of 1.7 – 3.0 and 1.6 – 7.5 respectively while the increase from 2011 – 2013 and an APC of 2.0, was not significant (CI -0.4 – 4.6). A test for parallelism of the two coverage rate trends for the on-time for MCV2 in the urban and rural locations to check for any difference between the two trend lines indicated no parallelism (p-value = 0.027) indicating a significant difference between the trend lines of the two geographical locations.

Page 13, lines 301 – 305: Summarizing, there was progressive increase in the coverage rates for both S-EDQ 1 and 5 and among both rural and urban populations for MCV2 in the province of Saskatchewan. The coverage rates were higher in the S-EDQ1 than S-EDQ5 and among the rural dwellers than the urban inhabitants among the on-time for MCV2 all through 2002 – 2013.

Page 24, line 568 – 570: It should be noted that data were not available for ethnicity and immigration status. Future studies are needed to explore patterns of measles immunization among other sub-populations which frequently exhibit inequities such as immigrants and indigenous population.

Results

R1.12: Lines 223-224. Are you referring to first or second dose coverage for measles, or full schedule coverage?

Authors’ response: We have included that second dose for measles (MCV2) was used which is also the full schedule according to World Health Organisation (WHO).

Change made: Page 11, line 250 – 251: From MCV2 results for 2002, Sun Country RHA had the highest coverage of 67.22% while Prairie North RHA had the lowest at 49.15%.

R1.13: Line 240-251. Specify what S-EDQ1 means. It is the first time that the abbreviation has been used but I cannot find what it corresponds to in the text. In addition to mentioning it in the Abbreviations section, it should also be included in the main text.

Authors’ response: In the edited document, the S-EDQ was first used on page 9, line 220 and the acronym has been fully specified. 

Changes made: Page 9, line 220: We referred to these quintiles as Socio-economic deprivation quintiles (S-EDQ1 – S-EDQ5) in this paper.

R1.14: Again, I would suggest including additional analysis on inequality in vaccination coverage, for example, differences by urban and rural sector, between migrants, between indigenous and non-indigenous populations. Particularly, in the qualitative part, the differences between the urban and rural sectors are mentioned; It would be interesting to see if these differences are evident in the quantitative part.

Authors’ response: We have included a section on rural and urban sector in the quantitative results section. 

Change made: Refer to R1.11 above.

Discussion

R.1.15: Line 430. What does HIT mean?

Authors’ response: HIT is an acronym for Herd Immunity Threshold. It was first mentioned in line 63. We have included the meaning of the acronym.

Change made: Page 3, line 61 – 63: Despite all the efforts to improve measles immunisation coverage, the proportion of immunised population with measles antigen containing vaccine (MCV) remains under the 95 percent herd immunity threshold (HIT).

R1.16: Line 433. To what period of time do these contextual adaptations refer (between 2002-2013? Or later). 

Author’s response: The contextual adaptations period spanned from 2002 to 2013, employed currently and in the future depending on needs.

Changes made: Page 20, lines 487 – 490: All these efforts are towards tailoring immunisation service to the targeted beneficiaries [63]. These approaches were employed in different combinations by RHAs in the past, currently and into the future with modifications depending on contexts.

R1.17: Since when did promoters of community programs begin to be introduced? Could it be observed if the presence of community program promoters really had an effect on vaccination coverage?

Authors’ response: Community program promoters were first introduced in 2007 to address the inequities in vaccination coverage from the inner-city core neighborhoods. It was effective in improving the coverage even though it was capital intensive. 

Change made: Page 20, lines 470 – 473: Particularly, regional health authorities where targeted approaches were introduced observed improvement in immunisation uptake by the initiation and maintenance of community program builders recruited from the community to reach out to the parents and clients in the inner-city core neighborhoods in Saskatoon RHA in 2007.

Page 20, lines 479 – 481: The above efforts are part of the strategies and interventions that increased uptake of measles immunisation especially by bridging the gap in the socio-economic divide with a resultant improvement in the coverage rates over the period under study.

R1.18: Line 448. What period of time do you refer to?

Authors’ response: We referred to the targeted approach to immunization period which started in Saskatchewan in 2007

Changes made: Page 20, lines 482 – 486: During the targeted approach period starting from 2007, many regional health authorities in the province of Saskatchewan utilized reminder systems more effectively, increased the number of immunisation service delivery points and became more flexible in terms of clinic opening hours to allow access for caregivers who could not attend within the strict regular opening hours

R1.19: Line 454 and following. I suggest, for the reader's clarity, to make a summary table of the changes over time in health strategies and policies related to vaccination, which are mentioned both in the results and in the discussion.

Authors’ response: The changes made varied in the respective RHAs. We have summarized the outcomes of the changes under the different themes in Fig 7 on page 19 of the manuscript.

Changes made: None

Figures.

R1.20: Figure 2. Include a note at the bottom of the figure that explains the graph and legends. Explain what S-EDQ is.

Author’s response: The legend can be found on the figures.

Changes made: None

R1.21: Figure 5. Include a footnote to the figure that explains the graph and captions. Explain what is DEP 1, 2 etc.

Author’s response: Footnote has been included. The DEP 1, 2 etc. are the same as S-EDQ 1, 2 etc. For uniformity and consistency, we have changed the DEP to S-EDQ on the graph.

Changes made: As indicated in the authors’ response above.

R1.22: Figure 6. Include title, explain the main elements of the figure in a footnote.

Author’s response: The title is indicated under the figure in the main document. The main elements were described in the main document. Figure 6 has become Figure 7 because of the inclusion of additional graph on rural and urban differences under results as part of the authors’ response to R1.11. 

Changes made: Page 18: lines 441 – 447: Figure 7 summarizes the results with various issues and clusters representing key building blocks—individual, social and institutional—that should be considered towards optimizing immunisation in the province. Each of these issues has critical implications for equity and public health policy. The framework includes individual, social and institutional dimensions of existing barriers and enablers and suggests a holistic, interconnected approach that integrates issues such as timeliness of measles immunisation dosages, population growth, urban-rural dichotomy, and regional differences and disparities in coverage rates.

---

## [Decision Letter · Decision Letter 1]

22 Aug 2022

PONE-D-21-15652R1Trends, Barriers and Enablers to Measles Immunization Coverage in Saskatchewan, Canada: a Mixed Methods StudyPLOS ONE

Dear Dr. Ilesanmi,

Thank you for submitting your manuscript to PLOS ONE. After careful consideration, we feel that it has merit but does not fully meet PLOS ONE’s publication criteria as it currently stands. Therefore, we invite you to submit a revised version of the manuscript that addresses the points raised during the review process.

We look forward to receiving your revised manuscript.

Kind regards,

Fernanda Penido Matozinhos, Ph.D

Academic Editor

PLOS ONE

Additional Editor Comments (if provided):

Dear authors,

Thank you for submitting your manuscript to PLOS ONE and making substantial changes in the manuscript.

After careful consideration, we feel that the goal of this study is relevant and it has technical rigor but does not fully meet PLOS ONE’s publication criteria as it currently stands. Therefore, we invite you to submit a revised version of the manuscript that addresses the points raised during the review process – specially in the Results’ section.

Kind regards,

Reviewers' comments:

Reviewer's Responses to Questions

**Comments to the Author**

1. If the authors have adequately addressed your comments raised in a previous round of review and you feel that this manuscript is now acceptable for publication, you may indicate that here to bypass the “Comments to the Author” section, enter your conflict of interest statement in the “Confidential to Editor” section, and submit your "Accept" recommendation.

Reviewer #1: All comments have been addressed

Reviewer #2: (No Response)

2. Is the manuscript technically sound, and do the data support the conclusions?

Reviewer #1: Yes

Reviewer #2: Partly

3. Has the statistical analysis been performed appropriately and rigorously? 

Reviewer #1: Yes

Reviewer #2: Yes

4. Have the authors made all data underlying the findings in their manuscript fully available?

Reviewer #1: Yes

Reviewer #2: (No Response)

5. Is the manuscript presented in an intelligible fashion and written in standard English?

Reviewer #1: Yes

Reviewer #2: Yes

6. Review Comments to the Author

Reviewer #1: I consider that the authors made the necessary changes and/or explanations required to guarantee the quality of the scientific article and corroborate the evidence presented.

Reviewer #2: Thanks to the authors for making substantial changes and revising the manuscript titled 'Trends, Barriers and Enablers to Measles Immunisation 1 Coverage in Saskatchewan Canada: a Mixed Methods Study'. This version looks much better. Some more edits and revisions are suggested below:

Please read and re-read the Abstract and for the full manuscript to fix the typos, and sentence construction errors. The reviewed pdf version with comments is attached herewith that'd aide authors on the specific corrections to be made.

Abstract: Please explain what constitutes 'social' and 'institutional' factors for immunization in the Methods section, and thereafter add this sentence

Keywords: Please delete repetitions - e.g. 'coverage'

Introduction: Please consider making this section more concise. Also in describing the measles vaccination scenario, introduce the global scenario, thereafter Canada, and lastly the study location. Currently, it is all over the place.

Methods and Results - Please have separate sections and subheadings for Quantitative and Qualitative when describing Data Source, and Data Analysis. Similarly in Results, report the Quantitative results first followed by the Qualitative Results.

Results: This section needs the maximum revision.

Please make the results section concise. Report only the findings here. Was any coding of the interviews undertaken to categorize the results under these 4 subheads? Access issues, fear issues, Anti vax issues, and systematic issues? If so, that needs to be elaborated, OR was this categorization adapted from any particular study or theoretical framework? If so that/those studies needs to be quoted.

More quotes are required to elaborate each section. Currently, the findings are mentions, without supporting quotes.

Discussion: Enhancement is needed in this section with addition of more citations

FOR COMMUNITY ENGAGEMENT

Dutta, T., Agley, J., Meyerson, B. E., Barnes, P. A., Sherwood-Laughlin, C., & Nicholson-Crotty, J. (2021). Perceived enablers and barriers of community engagement for vaccination in India: Using socioecological analysis. Plos one, 16(6), e0253318.

Dutta, T., Meyerson, B. E., Agley, J., Barnes, P. A., Sherwood-Laughlin, C., & Nicholson-Crotty, J. (2020). A qualitative analysis of vaccine decision makers’ conceptualization and fostering of ‘community engagement’in India. International journal for equity in health, 19(1), 1-14.

FOR CULTURE RESPONSIVE VACCINATION MESSAGING

Dutta, T., Agley, J., Lin, H. C., & Xiao, Y. (2021, May). Gender-responsive language in the National Policy Guidelines for Immunization in Kenya and changes in prevalence of tetanus vaccination among women, 2008–09 to 2014: A mixed methods study. In Women's Studies International Forum (Vol. 86, p. 102476). Pergamon.

7. PLOS authors have the option to publish the peer review history of their article (what does this mean?). If published, this will include your full peer review and any attached files.

Reviewer #1: No

Reviewer #2: **Yes: **Tapati Dutta

---

## [Author Response · Author response to Decision Letter 1]

24 Sep 2022

Authors’ response: Trends, Barriers and Enablers to Measles Immunization Coverage in Saskatchewan, Canada: a Mixed Methods Study

Abstract

E1: page 2, line 43. Review and revise this sentence. Deleting 'challenges' will make more sense

Authors response: Thank you for the edit. We have deleted “challenges” at the end of the end of the sentence as suggested.

Changes made: page 2, line 40 – 43: While access-related issues, caregivers’ fears, hesitancy, anti-vaccination challenges, and resource limitations were barriers to immunisation, improving community engagement, service delivery flexibility, targeted social responses and increasing media role were found useful to address the uptake of measles and other vaccine-preventable diseases immunisation 

E2: page 2 line 44-45. Please explain what constitutes 'social' and 'institutional' factors for immunization in the Methods section, and thereafter add this sentence

Authors response: We have explained the social and institutional factors for immunization in the Methods section.

Changes made: page 6, line 136 – 139. In this study, we describe the concept of social factors for immunization as that related to the conditions in which people grow, live and learn including culture and social norms, also work and group conformity [43] while institutional factors were looked at in form of situations, policies or procedures that systematically disadvantage certain groups of people. [44]

E3: page 2, line 51. Delete repetitions - 'coverage'

Authors response: Thank you for the edit. We have deleted coverage from Immunisation coverage.

Changes made: page 2, line 51: Measles; Immunisation; Coverage disparity; Health providers perspective; Equity; Joinpoint regression.

Introduction

E4: page 3, line 54 – 128. Please consider making this section more concise. Also introduce the global scenario, thereafter Canada, and lastly the study location. Currently, it is all over the place. 

Authors response: As recommended, we have tightened the introduction section in line with the request and made the section more concise while we addressed all the issues raised.

Changes made: page 3 – 7, line 53 – 123. Core to reducing the economic impact of diseases both on the health care financing and for children well-being is disease prevention achieved through immunisation, which has proved to be useful in the control and elimination of life-threatening infectious diseases [1, 2]. Among several infectious diseases, measles has received prominent attention internationally due to its high rate of infectivity [3-5]. Measles immunisation averts between 2 and 3 million deaths globally each year [2, 6]. Between 2000 and 2018, 23.2 million deaths were prevented [7] resulting in approximately 73% drop in measles cases within that period. With all the gains, however, in 2019, there were 869 770 measles cases worldwide, being the highest number since 1996 and with about 207 500 (23.9%) lost lives in the same year.[8] Despite all the efforts to improve measles immunisation coverage, the proportion of immunised population with measles antigen containing vaccine (MCV) remains under the 95 percent herd immunity threshold (HIT). [8] A high percentage of unimmunized individuals portends high susceptibility and infectivity for vulnerable groups with unplanned public health expenditure and negative outcomes [9]. 

Measles is a notifiable disease to be reported by health care providers in Canada as recommended in the national guideline since 1924.[12] The National Advisory Committee on Immunisation (NACI) emphasized the importance of elimination of indigenous measles since 1980, [13, 14] an objective which was reinforced by the Canadian Paediatric Association.[15] In 1994, Health Canada joined other Pan American Ministries of Health to set year 2000 measles elimination target in the Western Hemisphere [13, 17, 18] Measles elimination was however achieved in 1998 when Canada was declared free of endemic measles infection. [13, 18] Canada adopts 2-dose measles antigen-containing vaccination (MCV) recommendation by the World Health Organization (WHO),[19] however, with differential administration of the 2-dose approach where MCV1 is uniformly offered at 12 months, but varying timing of MCV2 across the ten provinces and three territories. Despite a strong institutional and organizational medical arrangement, immunisation is not mandatory at the national level in Canada. The provinces of Ontario, New Brunswick and Manitoba have implemented legislation that requires proof of immunisation for school entry [20] but immunisation, however, remains largely discretionary in other provinces. In many jurisdictions around the world, mandatory immunisation has been attempted for varying reasons with varying results, however, evidence did not support its effectiveness [21]. With the variations in the legal, policy and practice approach to immunisation, measles infection remains a source of concern in preventive medicine practice and research communities. Immunisation coverage rates measure the numbers of individuals who have received the appropriate doses by a specific date or age and are a reliable indicator of the preventative measures to control the spread of disease. Canada operates a goal of 95% coverage of one dose of measles immunisation by 2 years and 2 doses by 7 years [22]. The recent re-emergence of measles in some pockets of the population in Canada casts a doubt on possible elimination in the near term [23-25]. Some studies have linked the re-emergence with a low level of herd immunity threshold (HIT) for the disease. [26] In particular, low population coverage with the recommended 2-dose regimen for measles vaccine by age of 2 years has been flagged as a causal factor [27]. 

Research into equity gaps in childhood immunisations exists but very little is known in a Saskatchewan context. Differences in immunisation coverage within and between health regions or geographical areas have been documented in the literature [28-30] with differences in health arising from social determinants of health between different groups making it challenging for some individuals and groups to integrate fully in the society, which affect such individual’s health-seeking behaviour. Considering the established linkage between social determinants of health and health outcomes, [10, 11] public health institutions are aware of the effect of health inequalities and inequities; however, evidence is not rife on whether these efforts have had impact on equity gaps. Several factors influence the rate of immunisation like poor access, low education, limited family support and poverty, religious beliefs and colony formation [31, 32] among others. Multiple chains of transmission have been documented among religious communities that actively oppose or resist immunisation efforts. [13, 33-35] 

The province of Saskatchewan routine measles immunisation schedule are two doses of measles containing vaccine with first dose recommended at one year (12 months) and a second dose starting from 18 months of age. [16] Publicly available data show an unequal distribution of measles immunisation coverage among the health regions in the province of Saskatchewan. In 2014, the province recorded 75.3% at 2 years MCV1 coverage and 91% for MCV2 at 7 years age, [36] a range of 63.6% - 86.4% for MCV2 among regional health authorities in 2016, [37] and 79.9% average (69.1% - 93.2%) in 2018.[38] Evidence from the Saskatoon Health Region (SHR) suggests that incomplete immunisation with the attendant low coverages is primarily associated with low income, single parenthood, cultural status, and differences in beliefs [39], and where immunisation disparities exist between rural and urban areas and from neighbourhood to neighbourhood. 

Understanding the temporal trends in and drivers of measles immunisation coverage in small-area geographies of Saskatchewan can offer insight into how to improve immunisation policy development and implementation practices in the province of Saskatchewan and in contextually similar Canadian and international jurisdictions. Previous studies which looked at immunisation uptake in the province of Saskatchewan have examined the perspectives of the caregivers [30, 40, 41], but little is known on the perspectives of the health care providers as an important contributor to understanding coverage issues. Apart from the large regional health authorities such as Saskatoon and Regina Qu’Appelle Health Regions, which had carried out studies to document the coverage within their jurisdictions, there is scanty information on coverage rates at sub-regional levels, or within various quintiles of deprivation in the rest of the province. This study therefore examines measles immunisation coverage among health regions in the Canadian province of Saskatchewan and explores the barriers and enablers to achieving herd immunity threshold in the province and its smaller geographies. 

Materials and Methods

E5: page 6, line 141. Please have two separate sections Quantitative and Qualitative when describing Data Source, Data Analysis. Similarly in Results, report the Quantitative results first followed by the Qualitative Results.

Authors response: We have complied with your suggestion with clear subtitling of quantitative and qualitative sections under the Data Source, Analysis and the results sections

Changes made: 

Data source section: page 6, line 144. Quantitative Data. A total sample of 16,582 children on first dose (MCV1) ….’ Page 7, line 157. Qualitative Data. The qualitative part utilized interview data obtained from eighteen purposively…’

Data analysis section: page 8; line 187. Quantitative. The immunisation data was analysed using…’; 

page 9, line 221. Qualitative. The qualitative analysis employed hybrid…’

Results section: page 10 line 240. Quantitative Results; page 13, line 314. Qualitative Results

E6: page 7, line 156. Start this as a different section

Authors response: As recommended, we have made subtitle for the qualitative data collection methods and started in a new section. 

Changes made: page 7, line 157 – 159. Qualitative Data

The qualitative part utilized interview data obtained from eighteen purposively selected participants interviewed at two different times.

E7: page 7, line 166. Add space before parenthesis

Authors response: Space before parenthesis made

Changes made: page 7, line 168. Qualitative data triangulation [45] was ensured by interviewing two stakeholders…

E8: page 7, line 168. Not clear: This needs to go to the REsults Section

Authors response: We have moved the sentence to the results section and also addressed the separation of results to quantitative and qualitive subtitles as recommended under E5 above.

Changes made: page 13, line 314 - 318. Qualitative Results. Eighty-three percent (n=15) of the interview participants had at least eleven years’ experience, either in their present positions or in Saskatchewan RHAs in various public and population health capacities; with MHOs having an average of thirteen years’ experience and Coordinators/Front-line Immunisation Officers, an average of twenty years.

E9: page 7, line 172. pertained

Authors response: Thank you for that edit suggestion. The word has been put in past tense.

Changes made: page 7, line 171. The information asked from the participants pertained to their work in those periods when the quantitative data were collected; hence they were reflecting on their experience during the time data were collected, taking into account the past and future implications.

E10: page 7, line 176. Strikethrough text

Authors response: We removed the strikethrough text

Changes made: page 7, line 175.

Results.

E11: page 10, line 231. Again, Please make the results section concise. Report only your findings here. Delete this highlighted part 'To understand.....'

Authors response: We have deleted the redundant words and made the result section more concise reporting only results. We also removed the highlighted text in line 232.

Changes made: 

page 10, line 243 – 244. We assessed the two-year age group for the study sample demographics, there is a progressive increase of the study population from 2002 to 2009 reaching the highest figure of 15,189 in 2009 and decreased gradually to 14,106 by 2013 but without reaching the lowest figure of 13,273 of 2002 (Fig 2A) …’

Page 12, line 293 – 293. The relationship between the socio-economic status and measles coverage rates in the province was compared using the result of a Joinpoint regression which modeled coverage trends is shown in Fig 5….’ 

E12: page 13, line 306. Was any coding of the interviews undertaken to categorize them under these 4 subheads? Access issues, fear issues, Anti vax issues, and systematic issues? More quotes are required to elaborate each section

Authors response: There was coding of the interview transcripts to arrive at the categorization which resulted in the use of the subheads. We addressed this comment under methods section by improving the methods adding more content of the approach taken.

Changes made: page 9; line 222 – 238. The qualitative analysis employed hybrid inductive and deductive thematic analysis strategy described by Fereday & Muir-Cochrane [53]. This process included development of the codes, testing code reliability, summarizing the data and identifying the initial themes, code connection and theme identification and legitimization of the themes. The interview data were transcribed verbatim by a team of institution professionals. It was then read by the researcher thoroughly for immersive familiarization before coding. Nvivo version 12 [54] was used for the data analysis. Ideas in the interviews were first sorted into codes. The passages in the transcribed texts data were identified with respect to the concept being addressed by respondents and the relationship between the concepts followed by the categorization of the codes based on their similarities and relative differences and relationship to the research question and the comprehensive literature review. The categorized codes were then organized to subthemes and the themes informed the categories. We ensured that identified subthemes met recurrence and repetition criteria by ensuring a full understanding of the data to reduce biases and preconceived notions [55]. It was found that some sections or passages of some interview response fitted more than one theme and hence were coded in multiple ways depending on how many themes they fit into. The themes were refined after reviews by the research team as well as by presentation and discussing with the interview participants for legitimization of the coded themes and the categorization.

Results

E13: page 14, Line 325. What is hesitancy of caregivers? Please explain.

Authors response: We refer to vaccine hesitancy as a reluctance, delay in accepting or refusing vaccines despite vaccination services availability. We changed the subtitle to Fear and hesitancy issues and explained further in the text under the new subheading for clarity while including more quotes from the interviews data.

Changes made: 

page 15, line 343. Fear and hesitancy issues

Page 15, lines 344 – 356. Caregivers’ decisions on vaccination are complex. Factors like experiences, information, even emotions and trust influence their decisions. The reluctance, delay in acceptance and sometimes refusal of vaccines despite vaccine services availability is referred to as vaccine hesitancy [43, 56]. Personal factors such as fear, anxiety, and general skepticism of caregivers regarding the subject of immunisation was recurrent in the interviews, and all the participants responses centered on some caregivers not bringing their children for the required measles injections. One of the most mentioned is fear of needles according to one participant “We definitely have some that just have fears and hesitancies of needles” but also fears related to whether the measles vaccine causes autism or autism-related disorders, the possibility of reactions to the vaccine, distrust of the system, and other parental concerns. According to participants, some parents sometimes wondered ‘how many is enough or too much?’ among the immunisation schedules that children have to take at every milestone in their early years, especially when measles vaccine is given in combination with other antigens in MMR and MMRV. Such hesitations can hamper the optimization of coverage rates where lack of education and engagement is missing at the grassroots level.

E14: Line 335. Please add appropriate quotes to explain this subheading

Authors response: Appropriate quote was added to the text under the subheading and the text modified to accommodate the included quotes.

Changes made: page 15, line 358 – 373: Related to the parental fear factor mentioned above is the prevalence of anti-vaccination ideology among certain segments of the population. While fear and anxiety are dependent on individual characteristics and can be mitigated with public education and thorough engagement strategies, anti-vaccination ideologies take a philosophical objection to immunisation in general, and to multiple vaccines for infants specifically. According to some interviewees, who described some characteristics of the anti-vaxxers to the measles immunisation in Saskatchewan as follows ‘…of course, there are anti-vaxxers. They don’t have any immunizations. They don’t go into the system at all where their name could be, and they’re home-schoolers. So, they’re not going to go in the school system. They’re not in the government systems…’ Some participants also are sometimes influenced by unsubstantiated news in social media. Participants also mentioned the potential role of collective cultural beliefs where people in communal living arrangements or tightly organized religious, cultural, or social groups can exhibit “group-think” when anti-vaccination beliefs are promoted within the group. ‘…we do have some religious group and then have people that just have philosophical objections for whatever reason, things they’ve read from the internet or the hype from people…’. Participants identified several places in the province where local social or religious groups hold and promote anti-vaccine ideas where more targeted approaches to gaining trust may be required.

E15: line 372. Improved communication can be combined with Media use. Community Engagement can be a separate section. Once again, for every component reported, or most of it, there has to be an appropriate quote supporting the claim 

Authors response: Thank you for your suggestion. We have changed the subheading to “Improved communication and increasing role of the media” and moved information under the role of the media to add to the improved communication piece. We also made community engagement a separate subheading. More quotes were also added.

Changes made: page 17, line 404. Improved communication and increasing role of the media.

Page 17, line 405 – 433. Almost all interviewees stress the importance of multiple channels of communication and engagement that include a reminder system, interprofessional communication and active community and stakeholder engagement. Reminder methods such as telephone, letters, emails, cards and the CANImmunize app (formerly immunizeCA app) can be helpful for clients to track and keep their immunisation schedule. For example, one interviewee suggests “once the child was 13 months old and they hadn’t received their 1-year immunisation, they’d get a little card in the mail that reminded them they needed to book an appointment.” Many health regions also promote the use of the CANImmunize app, an interactive app supported by the Public Health Agency of Canada to provide information about vaccines and immunisation frequently asked questions. Regular communication among health workers was also seen as an essential enabler for bridging knowledge gaps and to develop coordinated responses to client needs in order to build confidence in health care providers and the overall health system. In order to develop adequate knowledge, which is a prerequisite to organizational competency, training and workforce development require collaborations with other health care workers outside of the RHA, including physicians and Nurse Practitioners. At the individual level, some interviewees reported that the “handouts and pamphlets” provided at maternity wards upon delivery of babies have been helpful to engage new caregivers on immunisation conversations and to address any questions they may have. ‘…We give handouts and discuss immunizations at the maternity visit program and home visit right after the baby is born’

While this is less frequent, some interviewees mentioned that the media has an important role to play in enabling measles and other forms of immunisation uptake in the province of Saskatchewan. “We used it [the media] sometime for influenza and we had several people line up to get flu vaccine, even though we weren’t appropriately staffed to manage the turnout, but we were happy to have the people... the media really does help”. This includes the use of radio and TV to educate the public to encourage parents to immunize their children, and reiterates the importance of not only getting immunised but the timeliness of the immunisation: “I think it would be great for the media to continue to supplement social media messages, but media does provide excellent service especially the radio stations,” Despite the identification of a clear role for traditional media such as radio and TV, only a few interviewees mentioned the important role the social media can play and it is in regard to the fact that “we have not explored the use of social media to their full potentials”.

Discussion

E16: page 18, line 439. Uptake and herd immunity might not be so simplistically and directly related, as mentioned here.

Authors response: We modified the sentence.

Changes made: page 20, line 472 – 474. In this paper, we present the measles immunisation coverage in the province of Saskatchewan focusing on the trends, barriers, and enablers to identify where actions are required for improved uptake which could move the province towards achievement of herd immunity.

E17: page 22, line 523. Would it be more appropriate to mention Culturally responsive vaccination messaging?

Authors response: Since the subheading also included issues of diverse context, we have modified the subheading to read Culturally responsive and context-specific vaccination messaging. 

Changes made: page 24, line 561. Culturally responsive and context-specific vaccination messaging

E18: Discussion: Enhancement is needed in this section with addition of more citations

Authors response: We have improved the section and included more citations from relevant work. Thank you for the recommended citations. We found one quite appropriate to add to the community engagement section while we included other relevant ones especially under the various discussion subtitles.

Changes made: Pages 20 – 24, Lines 472 – 578

In this paper, we present the measles immunisation coverage in the province of Saskatchewan focusing on the trends, barriers, and enablers to identify where actions are required for improved uptake which could move the province towards achievement of a herd immunity. The study used both quantitative and qualitative data to explore trends, patterns, and issues in the province and by extension to similar health jurisdictions in Canada and internationally. Figure 7 summarizes the results with various issues and clusters representing key building blocks—individual, social, and institutional—that should be considered towards optimizing immunisation in the province. Each of these issues has critical implications for equity and public health policy. The framework includes individual, social and institutional dimensions of existing barriers and enablers and suggests a holistic, interconnected approach that integrates issues such as timeliness of measles immunisation dosages, population growth, urban-rural dichotomy, and regional differences and disparities in coverage rates.

Data trends and regional disparities

The practice of immunisation trends monitoring is primarily reliant on coverage rates for its estimation and utilizes data which are routinely collected as a proportion of prescribed target populations [58-61]. It is information from such coverage rates that are needed for required vaccine-needs projections and immunisation program planning and implementation, including the evaluation of progress toward both national and international goals. [62] The coverage goal for childhood vaccines developed in 2017 was 95% at 24 months for all publicly funded vaccines in all Canadian provinces and territories [63]. This goal is consistent with the WHO, and 2012 Global Vaccine Action Plans which provided a framework for more equitable access to vaccines currently in use for the full benefits derivable from them [62, 64], while reflecting the Canadian context which has been adopted since 1996. The trend in measles immunisation coverage across the ten health regions increased steadily from 2002 – 2013. The increase in trend may be a response to all the various health region-specific innovations and program implementation practices during the study period. While this is a positive sign, the HIT has not yet been reached, and in order to reach it, regular reporting on coverage rate at small area level with targeted approaches to under-immunized sub-populations will be required.

Contextual adaption of interventions (incentives, targeted response)

Contextual adaptations were mentioned by interview participants as an enabler of immunisation coverage. Particularly, regional health authorities where targeted approaches are introduced observed improvement in immunisation uptake by the initiation and maintenance of community program builders recruited from the community to reach out to the parents and clients in the inner-city core neighborhoods in Saskatoon RHA in 2007. These targeted approaches are valued by individuals and communities, not only for measles immunisation effort but also for opportunities to address clients’ other needs such as the determinants of the clients’ overall well-being. [65] This finding suggests that interventions and implementation practices do not have to take the form of ‘one size fits all’; rather, the introduction of culturally relevant and contextually appropriate approaches may be more effective than a standardized, universal approach to client engagement and outreach. This approach is similar to what was described as proportionate universalism by Carey et al, 2015 [66] and Egan et al, 2016 [65]. The above efforts are part of the strategies and interventions that increased uptake of measles immunisation especially by bridging the gap in the socio-economic divide [65, 67] with a resultant improvement in the coverage rates over the period under study. These innovations and practices were confirmed by the key informant qualitative findings of this study. During the targeted approach period starting from 2007, many regional health authorities in the province of Saskatchewan utilized reminder systems more effectively, increased the number of immunisation service delivery points and became more flexible in terms of clinic opening hours to allow access for caregivers who could not attend within the strict regular opening hours. The reminder call systems have been used in other similar situations with improved coverage results. [68-70] All these efforts are towards tailoring immunisation service to the targeted beneficiaries [71]. These approaches were employed in different combinations by RHAs in the past, currently and into the future with modifications depending on contexts.[70]

Cross-regional knowledge-sharing, community engagement, and collaborations

Prior to the implementation of “one-health authority” in the province of Saskatchewan, each of the 13 health regions operated a more siloed approach in terms of operational framework and data management. Community engagement and outreaches were tailored to each region’s context and managed by health professionals assigned to manage immunisation programs. While some of the gaps associated with this approach may have been bridged, local adaptation of innovations to improve coverage rates according to local need is desirable but learning from what has worked in similar situations is best in speeding up the “diffusion of innovation”. Both quantitative and qualitative data show some regions have better coverage rates due to geographical advantages and innovative practices. For example, while Regina and Saskatoon RHAs saw a progressive increase in coverage rates throughout the period under study, Sun Country attained the highest coverage figure of 84.2% for the on-time for 2nd dose measles vaccine. Flexibility in service delivery, a needs-based approach that considered client socio-economic position, incentivization are some of the areas where cross-regional knowledge sharing will be valuable to practice. Proactive community engagement has been found effective at improving immunisation uptake by reducing negative group-think and decisions that lead to community resistance. [72, 73] It should also be noted that proactive engagement with clients and communities help health care workers to assess client attitudes that point to vaccine hesitancy with an opportunity for such to be addressed thereby reducing resistance. [72] While community engagement has been found to be a great tool to build relationships and trust to reduce vaccine hesitancy, this approach however, requires continuous collaboration among health professionals, local communities’ stakeholders, and decision-makers in planning, designing, and implementing initiatives that support such practices.

Prevalent institutional and policy issues

The study also found that institutional factors such as funding, staffing and database integration remain a barrier for measles immunisation uptake. This was particular the case for health regions covering rural, often-remote locations, which have transportation access challenges. In some cases, confusion about whether immunisation records of individuals in Indigenous communities are up-to-date due to lack of coordination and linkage between the federal and provincial immunisation registries exist. Increased funding could help bridge staffing and resource issues, and more easily encourage health care providers and individuals with policy roles to explore options for better coverage. The registry coordination challenge, however, is not new and has been raised in other similar studies, especially with respect to the lack of access to First Nations immunisation data to calculate provincial coverage rates [57, 74]. Moreover, due to the mobility of some First Nations people between on-reserve and off-reserve homes, tracking and communicating immunisation updates may be duplicated or sometimes omitted. It is therefore imperative that a national immunisation registry is created to facilitate greater transparency, coordination and consistency in tracking coverage rates in provinces and to develop a more accurate, timely and meaningful national outlook for measles immunisation coverage [57]. 

Culturally responsive and context-specific vaccination messaging

Social and community beliefs can discourage individuals from accessing measles immunisation, [75, 76] and health care providers do not have the authority and resources to address this challenge since immunisations are recommended and not mandated public health policy in many jurisdictions [77, 78]. The scale of hesitancy and resistance is more pronounced in some religious communities [79] but are also amplified by access to social media where ‘pseudo-science’ circulate more easily [80, 81]. Several factors, including conspiracy theories, general distrust, belief in alternatives, and concerns about safety have been identified as the drivers of the resurgence of vaccine hesitancy in recent past [43, 80, 82]. Hospital admission for measles can be as high as 47% among unimmunised individuals who were age-eligible for measles immunisation and up to 53% may develop acute respiratory failure [83]. Health care providers can only do so much as the legal environment allows, which suggests that persuasion and education remain the only tools to engage ‘anti-vaxxers’. Therefore, there is need for a more coordinated approach within the health community and across government social sectors to engage individuals and communities where objection to measles immunisation is still prevalent to increase the profile of credible sources of vaccine safety information. Interview responses have established the benefits of the traditional media (TV and Radio) as a major enabler, but gaps still exist on the use of social media. [84] The need for staff and client social media literacy skills and education therefore becomes more imperative and urgent to enable the public to decipher credible sources of information on measles immunisation.

---

## [Editor Report · Decision Letter 2]

6 Nov 2022

Trends, Barriers and Enablers to Measles Immunization Coverage in Saskatchewan, Canada: a Mixed Methods Study

PONE-D-21-15652R2

Dear authors, the manuscript explores a very important topic and it has technical rigor. 

Thank you for submitting your manuscript to PLOS ONE and making substantial changes in order to improve the manuscript. 

We’re pleased to inform you that your manuscript has been judged scientifically suitable for publication and will be formally accepted for publication once it meets all outstanding technical requirements.

Kind regards,

Fernanda Penido Matozinhos, Ph.D

Academic Editor

PLOS ONE

Additional Editor Comments (optional):

Dear authors, the manuscript explores a very important topic and it has technical rigor.

Thank you for submitting your manuscript to PLOS ONE and making substantial changes in order to improve the manuscript.

We recommend its publication.
---

## [Editor Report · Acceptance letter]

14 Nov 2022

PONE-D-21-15652R2 

Trends, Barriers and Enablers to Measles Immunisation Coverage in Saskatchewan, Canada: a Mixed Methods Study 

Dear Dr. Ilesanmi:

I'm pleased to inform you that your manuscript has been deemed suitable for publication in PLOS ONE. Congratulations! Your manuscript is now with our production department. 

Kind regards, 

on behalf of

Dr. Fernanda Penido Matozinhos 

Academic Editor

PLOS ONE